# A three-dimensional immunofluorescence atlas of the brain of the hackled-orb weaver spider, *Uloborus diversus*

Gregory Artiushin[1], Abel Corver[2,3,4], Andrew Gordus[1,4]*

[1]Department of Biology, Johns Hopkins University, Baltimore, United States; [2]Department of Biology, Lund University, Lund, Sweden; [3]Johns Hopkins Kavli Neuroscience Discovery Institute, Baltimore, United States; [4]Solomon H. Snyder Department of Neuroscience, Johns Hopkins University, Baltimore, United States

## eLife Assessment

This **valuable** study provides a 3D standardised anatomical atlas of the brain of an orb-weaving spider. The authors describe the brain's shape and its inner compartments-the neuropils-and add information on the distribution of a number of neuroactive substances such as neurotransmitters and neuropeptides. Through the use of histological and microscopy methods the authors provide a more complete view of an arachnid brain than previous studies and also presents **convincing** evidence about the organisation and homology of brain regions. The work will serve as a reference for future studies on spider brains and will enables comparisons of brain regions with insects so that the evolution of these structures can be inferred across arthropods.

**Abstract** Spider orb-web building is a captivating, rare example of animal construction, whose neural underpinnings remain undiscovered. We created a three-dimensional atlas for the hackled-orb weaver, *Uloborus diversus*, based on immunostaining for the presynaptic component, synapsin, in whole-mounted spider synganglia. Whereas spider neuroanatomy has thus far been most comprehensively studied in cursorial species, this optically sectioned atlas contributes a continuous, finely resolved model of the central nervous system of an orb-web building spider. Aligned to this volume, we examined the expression patterns of neuronal populations representing many of the classical neurotransmitters and neuromodulators (GABA, acetylcholine, serotonin, and octopamine/tyramine), as well as a subset of neuropeptides (allatostatin A, crustacean cardioactive peptide (CCAP), FMRFamide, proctolin) – detailing immunoreactivity in an unbiased fashion throughout the synganglion to reveal co-expression in known structures (such as the arcuate body), as well as novel neuropils not readily apparent in prior spider research, including the tonsillar neuropil as well as a potential protocerebral bridge. These structures provide targets for future functional studies, and taken together, could represent a spider equivalent of the central complex, contributing to behaviors such as web-building.

## Introduction

Brain atlases are essential tools for neuroscience in model organisms – ranging from neuropil annotations (*Jundi and Heinze, 2020*), to neuronal subtype and transcriptional expression pattern atlases (*Zhang et al., 2023*), to ultrastructural connectivity maps (*Cook et al., 2019*; *Chua et al., 2023*; *Dorkenwald et al., 2024*; *Yim et al., 2024*; *Cook et al., 2025*; *Verasztó et al., 2025*; *White et al., 1986*; *Winding et al., 2023*). In recent years, three-dimensional atlases of major neuropil structures have

*For correspondence:
agordus@jhu.edu

Competing interest: The authors declare that no competing interests exist.

**eLife digest** Spiders display fascinating behaviors, perhaps none more famous than web weaving. Yet little is known about the neural processes underlying this skill. A necessary step toward unravelling this enigma is the development of a three-dimensional brain atlas of a web-building spider.

To date, spider brain anatomy has been studied primarily in species that do not build webs. Moreover, most previous work involved thin tissue sections stained to highlight specific structures. These sliced views can often be discontinuous, making it difficult to relate information across different stains and brain regions.

To overcome these limitations, Artiushin, Corver and Gordus employed an antibody-based staining method that preserves intact brains for confocal microscopy. This approach enabled the creation of a detailed, three-dimensional anatomical map. The researchers further used a computational approach to align multiple scans into a single reference volume, allowing comparative overlays of different neuron classes. This strategy made it possible to identify areas known from other spiders, as well as new features that had not been previously documented.

Using this atlas, Artiushin et al. described new details of established brain regions – including neural circuits containing major neurotransmitters and neuromodulators. They also identified previously unknown brain structures, such as a region they termed the tonsillar neuropil, as well as a potential protocerebral bridge.

This work represents the first standardized brain atlas for any spider species. While earlier studies have imaged individual neuronal populations across disparate species and brain regions, this atlas provides the most comprehensive view to date within a single species. The newly identified brain regions offer promising candidates for future functional studies and may play roles in the intricate process of web building. More broadly, these findings advance our understanding of spider neuroanatomy and will be of interest to comparative and evolutionary studies of brain structure across animals.

A brain atlas provides a foundation for functional studies that link neuronal activity to behavior. Web building in spiders is a complex, innate form of animal construction that unfolds over multiple hours in distinct stages, making it a compelling model system for investigating the neural basis of extended, sequential behaviors.

also been created for non-canonical arthropod study species (*Brenneis, 2022*), including a number of insects (*Adden et al., 2020*; *Althaus et al., 2022*; *Dreyer et al., 2010*) and spiders (*Steinhoff et al., 2017*; *Steinhoff et al., 2024*).

The hackled-orb weaver spider, *Uloborus diversus* (*Eberhard, 1972*), is an emerging model system for the study of orb-web building in spiders (*Corver et al., 2021*; *Miller et al., 2022*), whose central nervous system has yet to be investigated. To date, the majority of studies of the spider central nervous system have been performed in one de facto model species, *Cupiennius salei*, a cursorial spider which hunts without building webs for prey capture (*Babu and Barth, 1984*). While isolated anatomical treatments exist for orb weavers and other web-based spiders (*Long, 2021*; *Hwang et al., 2015*; *Long, 2016*; *Moon and Tillinghast, 2013*; *Park et al., 2013*; *Rivera-Quiroz and Miller, 2022*; *Becherer and Schmid, 1999*; *Steinhoff et al., 2024*; *Wegerhoff and Breidbach, 1995*; *Weltzien and Barth, 1991*), the preponderance of *C. salei* literature is even starker when considering examinations beyond general neuronal stains, where *C. salei* is essentially the only spider species in which the expression pattern of more than a single neurotransmitter has been broadly mapped (*Becherer and Schmid, 1999*; *Fabian-Fine et al., 1999*; *Fabian-Fine et al., 2015*; *Fabian-Fine et al., 2017*; *Loesel et al., 2011*; *Schmid and Becherer, 1996*; *Schmid and Duncker, 1993*; *Senior et al., 2020*; *Seyfarth et al., 1990*; *Seyfarth et al., 1993*; *Tarr et al., 2019*). Furthermore, the current understanding of spider brain anatomy is substantially based on tissue slice analysis, which can provide exceptional detail and avoid damaging superficial brain structures, but has the disadvantage of being often limited in completeness by the planes which authors chose to exhibit.

Given that the substantial behavioral adaptation of web-building may be reflected in the presence of necessary brain structures and distinct underlying neuronal circuitry, an important step in understanding the basis of this behavior is to have a detailed, foundational architecture of a nervous system

which generates it. We created a three-dimensional immunofluorescence atlas of major neurotransmitter and neuromodulator populations for *U. diversus*, using whole-mounted synganglia. Using immunostaining against the presynaptic marker, synapsin, we assembled a standard, full volume of the *U. diversus* synganglion onto which specific neurosignaling molecule expression patterns were aligned. These include markers for classical neurotransmitters (GABA and acetylcholine), neuromodulators (dopamine, serotonin, and octopamine/tyramine), and several neuropeptides (allatostatin A, proctolin, CCAP, and FMRFamide). These volumes provide comprehensive and comparable detail throughout the synganglion, in both undifferentiated and established regions – such as the arcuate body, whose layers become distinguishable through the use of neurosignaling molecule co-stains. We further identify several previously undescribed neuropils in the supraesophageal ganglion, and the neuronal subtype populations whose specific expression demarcates them.

## Results

The central nervous system of spiders is distinctive among arthropods for its compressed nature. Residing within the prosoma, the central nervous system, or synganglion, as it has been called (*Steinhoff et al., 2017*), is comprised of two major divisions named in reference to the esophageal passage

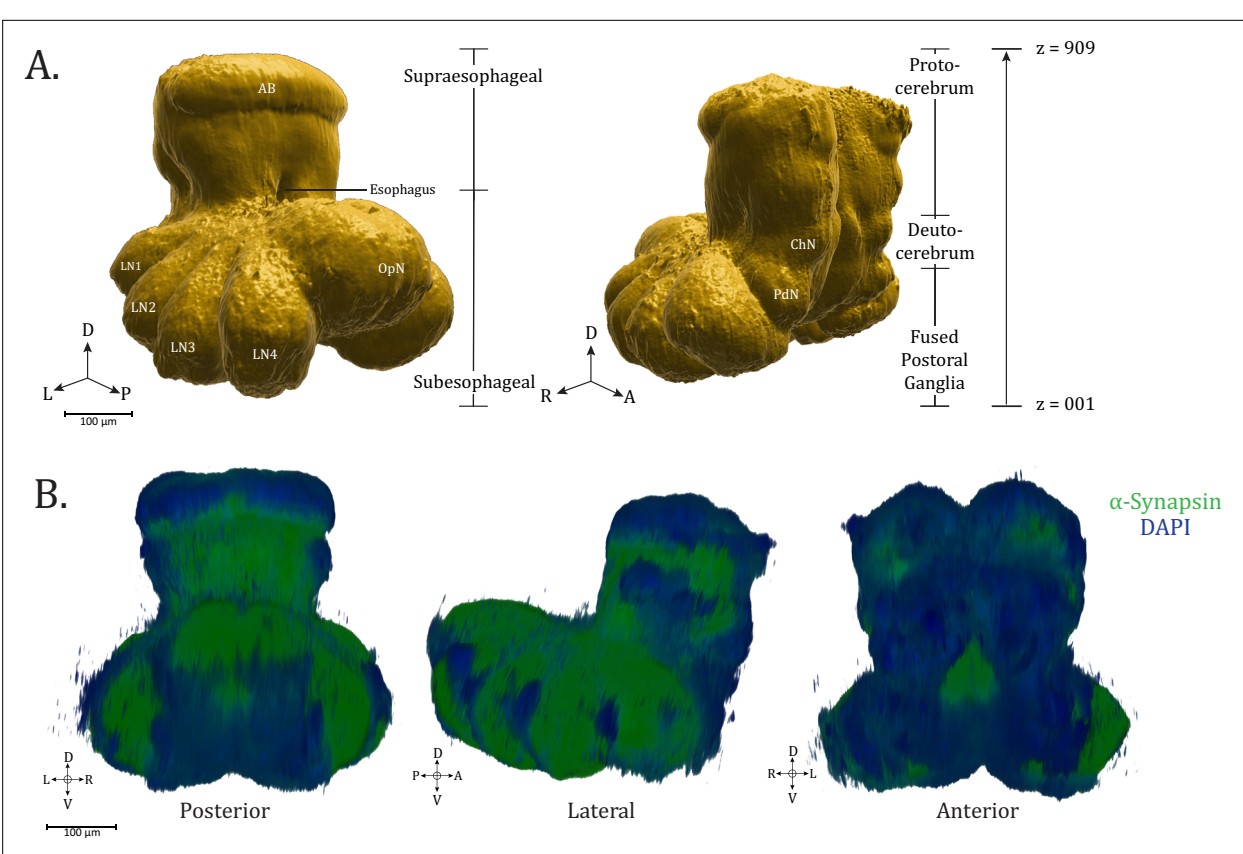

**Figure 1.** Synganglion of *Uloborus diversus*. (**A**) 3D rendering of *U. diversus* (female) synganglion from averaged α-synapsin volume, composed of 909 z-planes, oblique posterior–lateral (left) and oblique anterior–lateral (right) views. (**B**) 3D rendering of α-synapsin (green) and DAPI stained (blue) synganglion, posterior, lateral, and anterior views. Compass abbreviations: A = anterior, P = posterior, D = dorsal, V = ventral, L = left, R = right.

The online version of this article includes the following video(s) for figure 1:

**Figure 1—video 1.** α-Synapsin volume.

https://elifesciences.org/articles/107732/figures#fig1video1

**Figure 1—video 2.** Neurotransmitter volume.

https://elifesciences.org/articles/107732/figures#fig1video2

**Figure 1—video 3.** Neuropeptide volume.

https://elifesciences.org/articles/107732/figures#fig1video3

traveling between them – the subesophageal mass, comprised primarily of motor and sensory interneurons and comparable to the ventral nerve cord in insects, and the supraesophageal mass, containing the higher-order integration centers (*Figure 1A*). The divisions of the synganglion have also been described in regard to the dorsal-most protocerebrum (containing the optic, arcuate, and mushroom body [MB] neuropils), a deutocerebrum (comprised of the cheliceral neuropil [ChN] and esophagus), and the dorsal and posterior fused postoral ganglia (including the leg, pedipalp, and opisthosomal neuropils [OpN]) (*Long, 2021*; *Steinhoff et al., 2024*). Consistent with general arthropod nervous system morphology, the neuronal somata are found superficially (*Figure 1B*), while the internal structure of the brain is comprised primarily of neuropil.

To create a standard, three-dimensional brain atlas framework for *U. diversus*, we used elastix (*Klein et al., 2010*; *Shamonin, 2014*) to register and align multiple confocal image volumes of whole-mounted brain immunostains for the presynaptic marker, synapsin (3C11 antibody, DSHB). By averaging aligned image volumes (*n* = 6), we generated a consistent reference volume to use for further registration of more defined antibody targets, while revealing the patterns of neuropil structure of the *U. diversus* central nervous system, contiguously throughout the sub- and supraesophageal regions (*Figure 1—video 1*: synapsin), which we describe beginning at the ventral end and traveling dorsally.

Using anti-synapsin immunoreactivity as a common reference channel for all immunostains enabled application of the derived transformation matrix to respective co-stains for each brain sample, allowing incorporation of these channels into the standard brain, amounting to a total of eight neuro-signaling population immunostains in addition to anti-synapsin and DAPI (nuclear marker) unified in the current atlas. In this way, even though non-synapsin antibody targets were applied to separate brains, alignment of the synapsin channel to the reference enabled comparative analysis of co-expression of several antibody targets within the same reference volume.

We have collected standard brain-aligned confocal volumes for specific neurotransmitter and neuromodulator expressing populations, including cholinergic (anti-ChAT), serotonergic (anti-5-HT), and octopaminergic/tyraminergic (anti-TDC2) (*Figure 1—video 2*: neurotransmitters and neuromodulators). Neuropeptidergic expression patterns have been revealed for proctolin, allatostatin A, CCAP, and FMRFamide (*Figure 1—video 3*: neuropeptides) – whose functional significance remains unknown, but which have been previously imaged in spiders (*Loesel et al., 2011*; *Breidbach et al., 1995*). These aligned volumes have been made available via an online repository (see Data availability).

GABAergic expression pattern (anti-GAD) is aligned and reported, but of a more limited utility as the antibody signal penetration is limited to the periphery of the tissue. Certain target populations, such as dopaminergic neuron expression patterns (anti-tyrosine hydroxylase), are also provided, although we were not able to align the volume to the standard brain due to incompatibility of optimal staining conditions between the required antibodies. The use of other antisera, such as for β-Tubulin3 and horseradish peroxidase (HRP), is also demonstrated, but without alignment to the standard brain.

## Subesophageal (fused postoral ganglia) neuropils

### Leg neuropils

The leg neuropils (LNs) are the first and most apparent neuropil structures to appear in the ventral subesophageal mass and are present throughout most of the horizontal planes of this mass (until ~z400, standard brain) (*Figure 2A*; *Figure 2—video 1*; *Figure 2—video 2*; *Figure 2—video 3*). Their structure and innervation is generally consistent for all legs, with the only discernible difference being that LN1 are larger than those of the other legs, allowing for innervation patterns to be more easily characterized.

Most notably in the neuromeres of Legs I (anterior), the serotonergic (anti-5-HT) innervation in the limb neuroarchitecture appears to be supplied in two roughly equal halves, filling the periphery and leaving an area dark of immunoreactivity within (*Figure 2Bi* – dotted boundary). The anterior half of the innervation appears to be supplied from the medial branch of the 'dorsal-most tract' (as referenced by *Auletta et al., 2020*). Dopaminergic signal (anti-TH) is more expansive, evenly filling each LN with a mesh-like network of TH+ varicosities (*Figure 2D*). Unlike the approximately uniform innervation produced by dopaminergic neurons, the pattern in TDC2 staining is notably different. The anterior side of each neuromere contains a patch of continuous, diffuse, and more lightly stained immunoreactivity, while on each posterior side, there is a swath of brightly reactive, sparse puncta (*Figure 2Bii*). Cholinergic ChAT immunoreactivity is punctate and broadly filling of

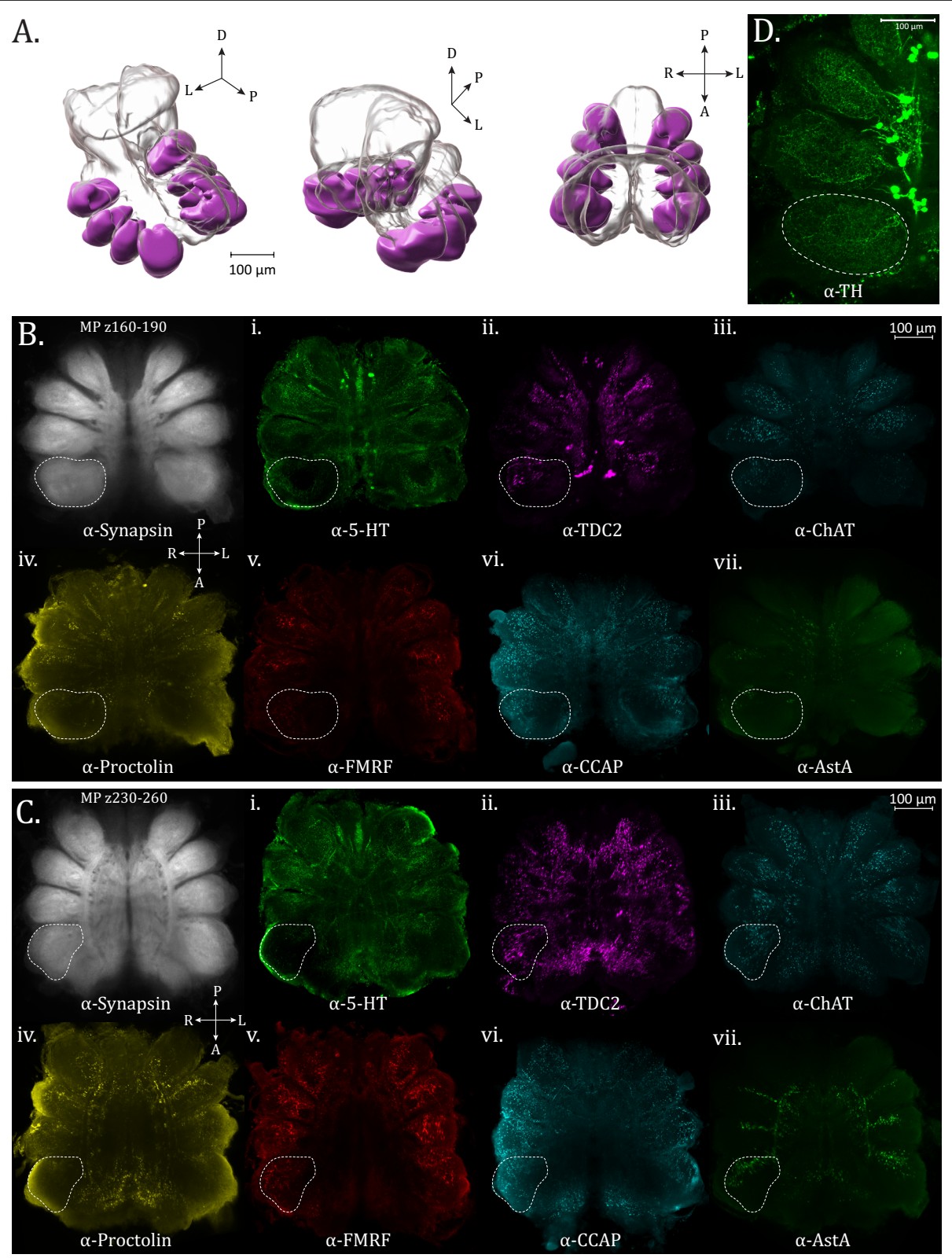

**Figure 2.** Leg neuropils. (**A**) 3D rendering of leg neuropils as annotated from averaged synapsin volume with posterior oblique, anterior oblique, and dorsal views, left to right. (**B**) Maximum intensity projections of z-planes 160–190 for leg neuropil expression of synapsin (α-synapsin, gray), (i) serotonergic (α-5-HT, green), (ii) octopaminergic/tyraminergic (α-TDC2, magenta), (iii) cholinergic (α-ChAT, cyan), (iv) proctolin (α-Proctolin, yellow), (**v**) FMRFamide (α-FMRFamide, red), (vi) cardioactive peptide (α-CCAP, cyan), and (vii) allatostatin A (α-AstA, green) immunoreactivity in the standard

*Figure 2 continued on next page*

*Figure 2 continued*

brain. Dotted perimeter marks the boundary of Leg Neuropil 1. (**C**) Maximum intensity projections of *z*-planes 230–260. Dotted perimeter marks the boundary of Leg Neuropil 1. (**D**) Maximum intensity projection showing dopaminergic (α-TH, green) immunoreactivity, dotted perimeter marks the boundary of Leg Neuropil 1. Dopaminergic immunoreactivity is shown in a separate panel because this volume, unlike the other stains, is not aligned to the standard atlas. Compass abbreviations: A = anterior, P = posterior, D = dorsal, V = ventral, L = left, R = right.

The online version of this article includes the following video(s) for figure 2:

**Figure 2—video 1.** Leg neuropil 3D volume.

https://elifesciences.org/articles/107732/figures#fig2video1

**Figure 2—video 2.** Neurotransmitter immunostains in the leg neuropil.

https://elifesciences.org/articles/107732/figures#fig2video2

**Figure 2—video 3.** Neuropeptide immunostains in the leg neuropil.

https://elifesciences.org/articles/107732/figures#fig2video3

the neuropil at the ventral end (*Figure 2Biii*) and more limited to the medial portion on the dorsal end (*Figure 2Ciii*).

AstA immunoreactivity has a distinctive pattern within the leg neuromeres, showing robust varicosities but only in the dorsal–posterior portion of neuropil (*Figure 2Cvii*). This innervation appears to be supplied from the lateral branches of the centro-lateral tract. A similar pattern regarding expression only in the posterior aspect of the neuropil is observed for proctolin immunoreactivity (*Figure 2Civ*). CCAP immunoreactivity within the LNs is predominantly in the posterior halves, where sparse puncta are evenly distributed (*Figure 2B, Cvi*). FMRFamide immunoreactivity is also evident in the LNs (*Figure 2B, Cv*).

## Opisthosomal neuropil

The OpN is found at the posterior and dorsal-most extension of the subesophageal mass (*Figure 3A*; *Figure 3—videos 1–3*). The region of the OpN has also been called the abdominal ganglia (*Foelix, 2025*), and tapers posteriorly to give way to the cauda equina, a nerve bundle composed of multiple paired tracts which innervate organs within the opisthosoma, such as the gonads, book lungs, and spinnerets (*Gonzalez-Fernandez and Sherman, 1984*). Within the OpN, there are a number of antero-posterior longitudinal tracts, while a ladder-like appearance of medio-laterally running tracts in the α-synapsin channel can also be observed (*Figure 3B, C*). Posteriorly traveling tracts also diverge laterally to follow the circumference of the OpN (*Figure 3B–D*).

Particular OpN features are revealed by a number of specific immunostains, including the monoamines octopamine/tyramine (TDC2 immunoreactivity), as well as neuropeptides such as proctolin and allatostatin A. TDC2 immunoreactivity displays an intricate pattern within the OpN. At the ventral anterior end, two triangular formations of puncta (*Figure 3Bii* – brace) abut the input of a string of varicosities on each lateral side, which then become heavier and continue to outline the boundary of the OpN (*Figure 3Bii* – arrow). The lateral perimeter tracts are likewise revealed by ChAT and proctolin immunoreactivity (*Figure 3B, Ciii, iv*) as well as by fibers from dopaminergic populations (anti-TH) (*Figure 3D*).

Within the interior, fibers resembling spokes emanate to a ring-like midline where there appears to be a small decussation and a thicker bridge structure joining lateral segments which travel in the anterior–posterior direction. The fine neurites projecting to the center of the OpN as seen for TDC2 immunoreactivity are also apparent for proctolin immunoreactivity (*Figure 3Bii, iv* – arrowhead). Proctolin signal also reveals a more posterior and dorsal crossing-over point (*Figure 3Civ* – arrow). Intense boutons line a tract running parallel to the midline, while also giving rise dorsally and laterally to a ladder-like structure of projections in the anterior–posterior direction, which can best be seen with octopaminergic/tyraminergic innervation (TDC2+) (*Figure 3Cii* – brace) and dopaminergic signal (TH+) (*Figure 3D*).

AstA+ immunoreactivity is also found in the OpN, with a more confined medial density in the dorso-posterior section, a pattern generally shared with 5HT and proctolin immunoreactivity (*Figure 3Cvii*). Ample FMRFamide signal is seen within the opisthosomal neuromere, but its signal is more uniform than the other targets investigated (*Figure 3B, Cv*).

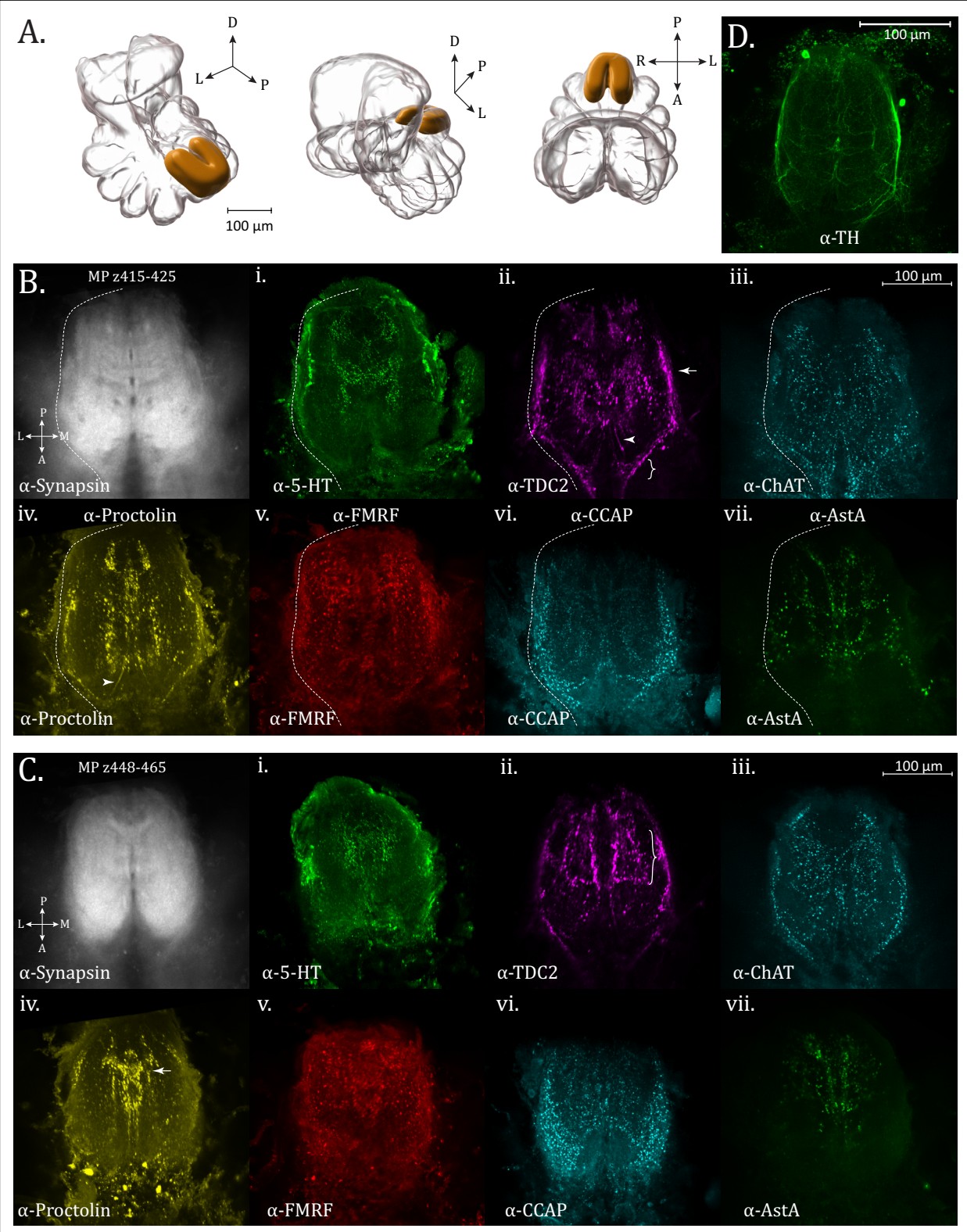

**Figure 3.** Opisthosomal neuropil. (**A**) 3D rendering of the opisthosomal neuropil as annotated from averaged synapsin volume with posterior oblique, anterior oblique, and dorsal views, left to right. (**B**) Maximum intensity projections of z-planes 415–425 for opisthosomal neuropil expression of synapsin (α-synapsin, gray), (**i**) serotonergic (α-5-HT, green), (**ii**) octopaminergic/tyraminergic (α-TDC2, magenta), (**iii**) cholinergic (α-ChAT, cyan), (**iv**) proctolin (α-Proctolin, yellow), (**v**) FMRFamide (α-FMRFamide, red), (**vi**) cardioactive peptide (α-CCAP, cyan), and (**vii**) allatostatin A (α-AstA, green) immunoreactivity in the standard brain. Arrow in the α-TDC2 subpanel marks tracks along the perimeter of the opisthosomal neuropil, supplied

*Figure 3 continued on next page*

*Figure 3 continued*

anteriorly (brace). Arrowheads in α-TDC2 and α-Proctolin subpanels mark a neurite running centrally. (**C**) Maximum intensity projections of *z*-planes 448–465. Longitudinal and lateral projections form a ladder-like structure in the α-TDC2 subpanel (brace). A midline-crossing structure (arrow) is seen in the α-Proctolin subpanel. (**D**) Maximum intensity projection showing dopaminergic (α-TH, green) immunoreactivity in the opisthosomal neuropil. Compass abbreviations: A = anterior, P = posterior, D = dorsal, V = ventral, L = left, R = right.

The online version of this article includes the following video(s) for figure 3:

**Figure 3—video 1.** Opisthosomal 3D volume.

https://elifesciences.org/articles/107732/figures#fig3video1

**Figure 3—video 2.** Neurotransmitter immunostains in the opisthosomal neuropil.

https://elifesciences.org/articles/107732/figures#fig3video2

**Figure 3—video 3.** Neuropeptide immunostains in the opisthosomal neuropil.

https://elifesciences.org/articles/107732/figures#fig3video3

## Pedipalpal neuropil

As the LNs begin to diminish in the dorsal region of the subesophageal mass, a smaller antero-lateral synaptic density becomes apparent, representing the pedipalpal neuropil (PdN) (*Figure 4A*; *Figure 4—videos 1 and 2*), which serves the anterior-facing pedipalp appendages. Immunoreactivity for ChAT (acetylcholine) and TDC2 (octopamine/tyramine) is strongest among the antisera tested for the PdN, showing punctate expression which does not extend to fill the anterior portion of the neuropil (*Figure 4Biii and ii*, respectively). No additional structural features were evident within the PdN. Little to no appreciable immunoreactivity was seen for allatostatin A or CCAP antisera in our defined bounds of the PdN. This was also true of proctolin immunoreactivity, although signal is present in an immediately medial, adjacent region (*Figure 4Biv*), which is also highlighted by 5-HT immunoreactivity (see z275 in atlas). This area appears to be supplied by at least two anteriorly located proctolin+ somata (*Figure 4Biv* – arrow).

## Blumenthal neuropil

An additional subesophageal feature previously identified in *C. salei* is the Blumenthal neuropil (*Anton and Tichy, 1994*), which is innervated by afferents from the thermoreceptive and hygroreceptive tarsal organ. Although we also see a paired, synapsin-dense entity close to the midline (*Figure 4C* – brace), in the approximate anterio-ventral subesophageal location as described for *C. salei*, we cannot be certain that this is the same structure – a question which will benefit from tracing techniques.

A circular form of saturated proctolin immunoreactivity is seen at the posterior end of an oval-shaped synapsin density (*Figure 4Ci, ii* – dashed perimeter), suggesting that it is a subset of a major tract bundle. In dorsal planes, this immunoreactivity morphs into laterally moving tracts *Figure 4—video 6*, becoming difficult to trace. Immunoreactivity within this synapsin-dense perimeter (*Figure 4Ci*) is not found in the other neurosignaling molecule stains, even those with otherwise abundant subesophageal expression.

## Deutocerebral features
### Cheliceral neuropil

At the level of the esophageal passage, an anterior–lateral neuropil begins, wrapping medially to become the ChN (*Figure 4D*; *Figure 4—videos 3–5*). This neuropil is associated with the fanged appendages known as chelicerae, which are used for prey handling and feeding. Anterograde tracing of the innervation pattern of lyriform organ and tactile hair mechanosensors of the chelicerae of *C. salei* found projections to terminate in the ChN as well as dorso-ventrally in the sensory longitudinal tracts of the subesophageal mass (*Gorb et al., 1993*). The ChN is most abundantly innervated by serotonergic as well as TDC2+ immunoreactivity (*Figure 4Eiv, v*). Fine varicosities of 5-HT+ immunoreactivity fill the neuropil, while the substantial TDC2+ immunoreactivity appears as large puncta throughout. TDC2+ expression is also strong in a region immediately medial to the ChN, where 5HT+ immunoreactivity is also found (*Figure 4Ev* – arrows). While AstA+ signal is marginal within the demarcated ChN, strong immunoreactivity is also evident in the adjacent medial region, which overlaps with TDC2+ and 5HT+ expression (*Figure 4Eiii* – arrows). ChAT (cholinergic immunoreactivity) is also

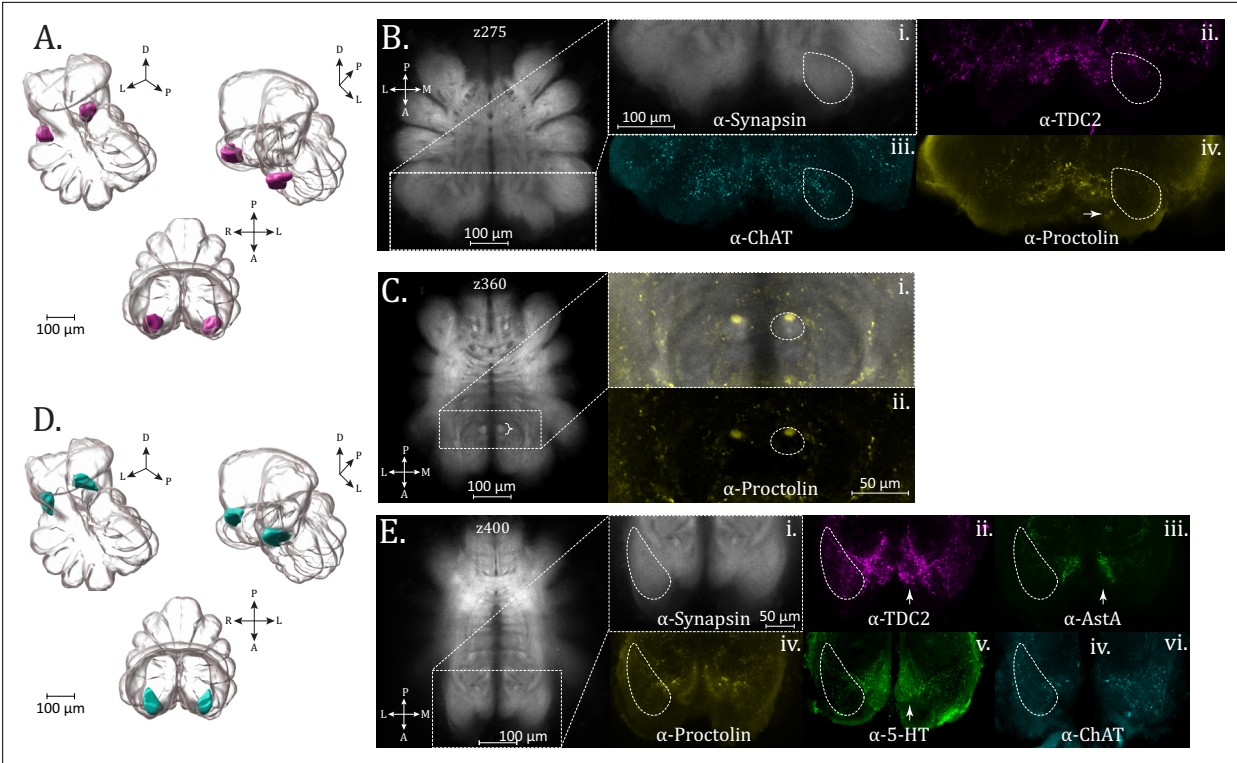

**Figure 4.** Pedipalp, Blumenthal, and cheliceral neuropil. (**A**) 3D rendering of the pedipalp neuropil as annotated from averaged synapsin volume with posterior oblique and anterior oblique views, left to right, and dorsal view, lower. (**B**) Optical slices (z275) from the standard brain, containing a cross-section of the pedipalp neuropil (dashed line boundary), with expression of (i) synapsin (α-synapsin, gray), (ii) octopaminergic/tyraminergic (α-TDC2, magenta), (iii) cholinergic (α-ChAT, cyan), and (iv) proctolin (α-Proctolin, yellow) immunoreactivity. Arrow in α-Proctolin shows adjacent proctolin+ somata. (**C**) Optical slices (z360) from the standard brain, with a cropped and enlarged selection showing the Blumenthal neuropil (brace), with expression of (i) synapsin (α-synapsin, gray) and proctolin (α-Proctolin, yellow) immunoreactivity above, and only (ii) proctolin below. A circular region of the Blumenthal neuropil (dashed oval) shows proctolin immunoreactivity. (**D**) 3D rendering of the cheliceral neuropil as annotated from averaged synapsin volume with posterior oblique and anterior oblique views, left to right, and dorsal view, lower. (**E**) Optical slices (z400) from the standard brain, containing a cross-section of the cheliceral neuropil (dashed line boundary), with expression of (i) synapsin (α-synapsin, gray), (ii) octopaminergic/tyraminergic (α-TDC2, magenta), (iii) allatostatin A (α-AstA, green), (iv) proctolin (α-Proctolin, yellow), (v) serotonergic (α-5-HT, green), and (vi) cholinergic (α-ChAT, cyan) immunoreactivity. Arrows in α-TDC2, α-AstA, and α-5-HT subpanels mark an unidentified medially adjacent region to the cheliceral neuropil which shows pronounced immunoreactivity for these antisera. Compass abbreviations: A = anterior, P = posterior, D = dorsal, V = ventral, L = left, R = right.

The online version of this article includes the following video(s) for figure 4:

**Figure 4—video 1.** Pedipalp neuropil 3D volume.

https://elifesciences.org/articles/107732/figures#fig4video1

**Figure 4—video 2.** Neurotransmitter and neuropeptide immunostains in the pedipalp neuropil.

https://elifesciences.org/articles/107732/figures#fig4video2

**Figure 4—video 3.** Cheliceral 3D volume.

https://elifesciences.org/articles/107732/figures#fig4video3

**Figure 4—video 4.** Neurotransmitter immunostains in the cheliceral neuropil.

https://elifesciences.org/articles/107732/figures#fig4video4

**Figure 4—video 5.** Neuropeptide immunostains in the cheliceral neuropil.

https://elifesciences.org/articles/107732/figures#fig4video5

**Figure 4—video 6.** Immunostains for α-synapsin and α-proctolin in the blumenthal neuropil.

https://elifesciences.org/articles/107732/figures#fig4video6

present throughout the ChN, while proctolin immunoreactivity is more minor, in the middle and anterior reaches of the structure (*Figure 4Evi, iv*).

## Supraesophageal (Protocerebral) neuropils

### Ventral features

The supraesophageal mass begins dorsally to the closure of the esophageal passage, and in its ventral-most planes, can be subdivided anteriorly and posteriorly into two sectors of slightly differing anti-synapsin immunoreactivity (*Figure 5A–C* – dashed perimeter, anti-synapsin). We will refer to these regions as the **anterior and posterior stalk**, in order to enable discussion of smaller features of various specific neurostains found within their perimeter. Lacking any further knowledge, these names are not currently intended to suggest a cohesive form or function for the structures within their bounds.

The esophageal passage is bridged at the anterior side by a region named the stomodeal bridge (StB) (*Steinhoff et al., 2017*; *Figure 5A* – brace). A bridge structure also exists at the posterior end, where additional undifferentiated synaptic density is flanking (*Figure 5B* – brace with asterisk). Within these planes, a protocerebral tract (PCT) is essentially parallel to the ventro-dorsal axis and appears as twin, dense nodes rising in the central burgeoning protocerebrum (*Figure 5B, C*).

5-HT immunoreactivity is prominent in the posterior bridging area dorsal to the esophageal passage (*Figure 5B, Ci*, *Figure 5—video 1*), as well as the laterally adjacent tissue, and not as apparent in the anterior StB. 5-HT immunoreactivity is otherwise weak within the stalk regions. TDC2 immunoreactivity is prominent in the StB and adjacent areas, and at this plane, two lateral bands of immunoreactivity appear which do not correspond to a clear demarcation in the synapsin channel (*Figure 5Biii* – arrow, *Figure 5—video 1*).

Dorsally, octopaminergic/tyraminergic (TDC2+) signal is prominent along an antero-lateral stretch marking the boundary of what we define as the posterior stalk, a perimeter also visible in the synapsin channel (*Figure 5Ciii*). Concentrations of TDC2+ immunoreactivity are also apparent centrally, both anterior and lateral to the PCTs (*Figure 5Ciii* – arrows, *Figure 5—video 1*).

Similar to what has been described for *M. muscosa* as the StB (*Steinhoff et al., 2017*), the area adjacent to the esophagus on the anterior side has immunoreactivity to allatostatin A, although the actual bridge which crosses the midline is modest, with thin representation in the posterior commissure (*Figure 5Av* – arrowhead, *Figure 5—video 2*).

In *U. diversus*, strong AstA immunoreactivity is present on the posterior side of where the esophagus closes, in the posterior stalk area (*Figure 5Cv*). The posterior region adjacent to the midline, previously highlighted with 5HT immunoreactivity, also shows partial AstA+ innervation, displaying a unique oxbow type pattern (*Figure 5Cv* – arrow, *Figure 5—video 2*). There is also a patch of AstA+ immunoreactivity in the central area of the anterior stalk (*Figure 5Cv* – brace).

On the posterior edge of the StB, there is a thin Proc+ commissure, while the anterior edge of the StB is highlighted by a bolder vein of varicosities (*Figure 5Aiv* – arrows). Proctolin signal is present adjacent to the midline around the posterior bridging area, shared with 5-HT, AstA, and TDC2+ immunoreactivity, and together with the commissure evident by synapsin staining, forms a circular pattern (*Figure 5Biv* – brace). This circular pattern of immunoreactivity is also visible by 5-HT+ immunoreactivity (*Figure 5Bi*), as well as TDC2+ innervation, though more clearly seen further dorsally (*Figure 5Ciii*) for this channel. There is also proctolin immunoreactivity seen centrally, medial to where the PCT is ascending (*Figure 5B, Civ*).

ChAT+ immunoreactivity is diffusely present throughout the stalk regions (*Figure 5Cii*), and unlike the other antisera, co-stains the posterior aspect of the PCT, which is prominently visible with synapsin immunoreactivity (*Figure 5B, Cii* – arrowheads and dashed circles).

## Hagstone neuropil and mid-supraesophageal features

We will define the hagstone neuropil as a central, midline adjacent, paired structure, whose ventral bounds share the same plane as the appearance of a prominent protocerebral commissure (standard brain, ~$z$ = 540), and which continues dorsally until the formation of the MB bridge (standard brain, ~$z$ = 615). This approximately kidney-bean-shaped structure is pierced by a circular spot lacking synapsin immunoreactivity (*Figure 6A, C, D* – dashed outline, *Figure 6—videos 1–3*) – which may be indicative of a space occupied by a fiber tract, tracheal passageway, and/or potentially glia.

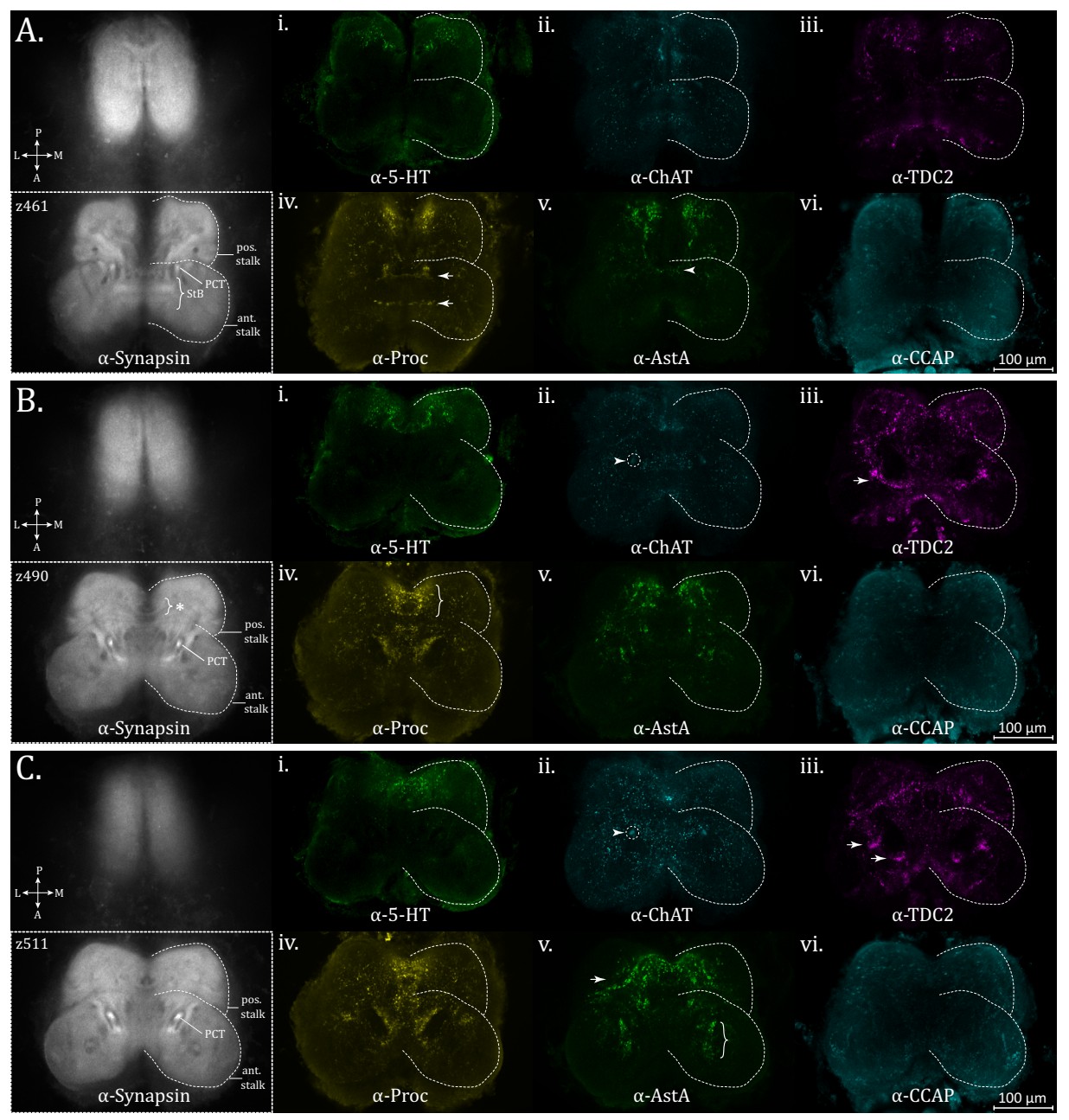

**Figure 5.** Ventral supraesophageal features. Posterior and anterior stalk region expression of synapsin (α-synapsin, gray), (**i**) serotonergic (α-5-HT, green), (ii) cholinergic (α-ChAT, cyan), (iii) octopaminergic/tyraminergic (α-TDC2, magenta), (iv) proctolin (α-Proctolin, yellow), (**v**) allatostatin A (α-AstA, green), and (vi) cardioactive peptide (α-CCAP, cyan) immunoreactivity in the standard brain, for: (**A**) z-plane 461. Arrow in the α-TDC2 subpanel marks tracks along perimeter of the opisthosomal neuropil, supplied anteriorly (brace). Top and bottom arrows in α-Proctolin subpanel represent posterior and anterior bridging immunoreactivity, respectively, in the stomodeal bridge (StB) area. Arrowhead in α-AstA subpanel marks a faint band of immunoreactivity across the midline. (**B**) z-plane 490. Further ventrally, a bridging structure is also visible on the posterior end of the ventral supraesophageal, as seen in the α-Proctolin subpanel (brace). Cholinergic immunoreactivity is present in the protocerebral tract (PCT), arrowhead in α-ChAT subpanel. Bands of α-TDC2 immunoreactivity (arrow) which do not correspond to clear structures in the synapsin channel. (**C**) z-plane 511. Arrowhead in the α-ChAT subpanel marks prominent cholinergic immunoreactivity in the PCT, more clearly visible at this plane. In the α-AstA subpanel, an arrow shows an oxbow-like structure and pronounced innervation at the posterior midline and central anterior stalk AstA+ innervation (brace). Arrows in the α-TDC2 subpanel mark centrally located concentrations of TDC2 immunoreactivity anterior and lateral to the PCT. Compass abbreviations: A = anterior, P = posterior, D = dorsal, V = ventral, L = left, R = right.

The online version of this article includes the following video(s) for figure 5:

*Figure 5 continued on next page*

*Figure 5 continued*

**Figure 5—video 1.** Neurotransmitter Immunostains in the Ventral Supraesophageal Ganglion.
https://elifesciences.org/articles/107732/figures#fig5video1

**Figure 5—video 2.** Neuropeptide immunostains in the ventral supraesophageal ganglion.
https://elifesciences.org/articles/107732/figures#fig5video2

The hagstone neuropil, as well as other prominent features of the mid-central protocerebrum, are most clearly defined by serotonergic immunoreactivity. Serotonergic fibers form a distinctive circular tract pattern around the midline of the supraesophageal ganglion (*Figure 6Bi* – brace), not as clearly seen with any other neuronal-subtype stain. The semi-circular tracts bend medially (*Figure 6Bi*), before a fusion of seemingly all three directions is seen immediately dorsal (*Figure 6C*). The hagstone neuropil is essentially completely filled with serotonergic immunoreactivity, matching the outline defined in the anti-synapsin channel (*Figure 6C, Di*).

On the posterior end is a diffuse, arching band of varicosities, with fiber tracts at the midline, resembling an umbrella-like form (*Figure 6C, Di* – brace, anti-5-HT). While the umbrella-like band formation is not distinct enough to be reliably annotated from synapsin immunoreactivity, we find that signal within this formation is also visible in other neuromodulator and neuropeptide immunostains.

TDC2 immunoreactivity is also present in the umbrella-like posterior region innervated by 5-HT (*Figure 6Ciii* – brace, *Figure 6—video 2*), and sparser puncta within the bounds of the hagstone neuropil. Proctolin immunoreactivity is present in the posterior, midline-spanning umbrella structure observed for 5-HT and TDC2, as well as fine varicosities in the hagstone neuropil (*Figure 6Dii*). This is likewise the case for AstA immunoreactivity in the hagstone neuropil, while posteriorly, there is abundant AstA+ signal which partially is coincident with the umbrella-like band. Posterior to the cup-shaped synaptic density formed by the MB hafts continuing with the rest of the MB, is a crescent of proctolin immunoreactivity (*Figure 6Cii* – brace), which also appears to be present in *C. salei* (*Becherer and Schmid, 1999*). ChAT immunoreactivity, as before, is found broadly, including within the hagstone neuropil, though no greater structure is discernible from this channel in the mid-supra otherwise. It is difficult to ascertain to what degree the innervation in this region is continuous with that of dorsally located features.

## Mushroom bodies

The mushroom bodies (MBs) of *U. diversus* (*Figure 7A, B*; *Figure 7—video 1*) tend to show the most robust synapsin immunoreactivity of all structures in the supraesophageal mass (*Figure 7B*, maximum intensity projection), indicating a great degree of synaptic density. While web-building species have been reported to have simplified (*Steinhoff et al., 2024*) or even entirely absent mushroom bodies (*Hanström, 1928*; *Long, 2016*; *Long, 2021*; *Steinhoff et al., 2024*) although notable exceptions have also been observed such as *D. spinosa* and *A. trifasciata* (*Long, 2016*), these structures are present in *U. diversus* and retain the complete form seen in more visually reliant species (*Steinhoff et al., 2017*; *Steinhoff et al., 2024*), even if they are smaller relative to the protocerebrum as a whole (*Figure 7A–C*).

*U. diversus* MBs display a haft, body, and head region, with the two pairs connected by a bridge (*Figure 7A–C*). Synapsin immunoreactivity is modest within the bridge region, whose true thickness is better visualized with staining for βTubulin3 (*Figure 7D*). Despite the strong synapsin immunoreactivity in the MBs, we surprisingly did not see co-expression with most of our specific neurosignaling molecule antibodies. This pattern is also reflected in the extant spider literature, with a single study showing immunoreactivity in the mushroom bodies of *C. salei* for anti-GAD and anti-proctolin staining (*Becherer and Schmid, 1999*). A particularly intensely stained MB bridge was also seen in *A. sclopetarius* using a non-antisera stain of acetylcholinesterase activity (*Meyer and Pospiech, 1977*). In our hands, only anti-allatostatin A staining showed co-expression throughout the MB (*Figure 7E*). Although difficult to trace the source, it appears the hafts are innervated from the posterior side (*Figure 7F* – arrows). By βTubulin3 immunoreactivity, we observe two tracts which straddle the MB hafts as they descend from the dorsal somata layer (*Figure 7G*). Finer neurites are not distinguishable in the βTub3 immunoreactivity, but it is plausible that the AstA+ neurites entering the MB hafts might stem from the medial of these two tracts.

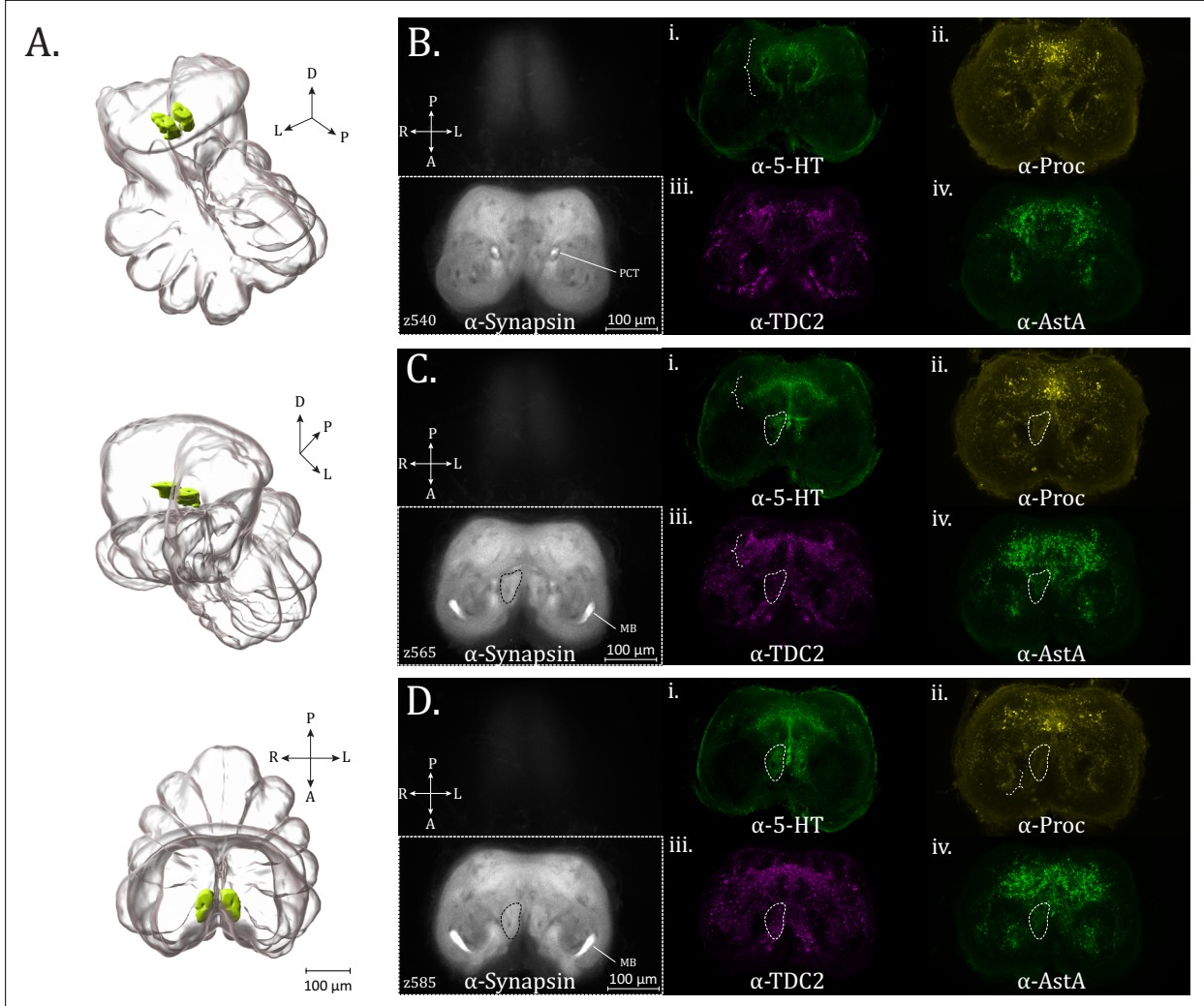

**Figure 6.** Hagstone neuropil and mid-supraesophageal features. (**A**) 3D rendering of the hagstone neuropil as annotated from averaged synapsin volume with posterior oblique, anterior oblique, and dorsal views, top to bottom. (**B–D**) Hagstone neuropil and posterior feature expression of synapsin (α-synapsin, gray), (**i**) serotonergic (α-5-HT, green), (**ii**) proctolin (α-Proctolin, yellow), (**iii**) octopaminergic/tyraminergic (α-TDC2, magenta), and (**iv**) allatostatin A (α-AstA, green) immunoreactivity in the standard brain, fo: (**B**) z-plane 540. A circular tract pattern distinctive to the α-5-HT stain is visible in the posterior mid-supraesophageal (brace, α-5-HT subpanel). (**C**) z-planes 565, with the boundary of hagstone neuropil (dashed perimeter). An umbrella-like innervation pattern is present posteriorly (brace), in both the α-5-HT and α-TDC2 subpanels. (**D**) z-plane 585, with the boundary of hagstone neuropil (dashed perimeter). In the α-Proctolin subpanel, a brace marks a crescent-shaped zone of Proctolin immunoreactivity, which has also been observed in another spider species. PCT = protocerebral tract, MB = mushroom body. Compass abbreviations: A = anterior, P = posterior, D = dorsal, V = ventral, L = left, R = right.

The online version of this article includes the following video(s) for figure 6:

**Figure 6—video 1.** Hagstone 3D volume.

https://elifesciences.org/articles/107732/figures#fig6video1

**Figure 6—video 2.** Neurotransmitter immunostains in the hagstone neuropil.

https://elifesciences.org/articles/107732/figures#fig6video2

**Figure 6—video 3.** Neuropeptide immunostains in the hagstone neuropil.

https://elifesciences.org/articles/107732/figures#fig6video3

*Babu and Barth, 1984* described the protocerebro-dorsal tract as providing input to the hafts of the mushroom bodies. The connection of this tract to the MB hafts is not apparent by synapsin or βTubulin3 immunoreactivity in *U. diversus*, which was likewise the case with silver staining for *P. amentata*, *M. muscosa*, *A. bruennichi*, and *P. tepidariorum* (*Steinhoff et al., 2024*).

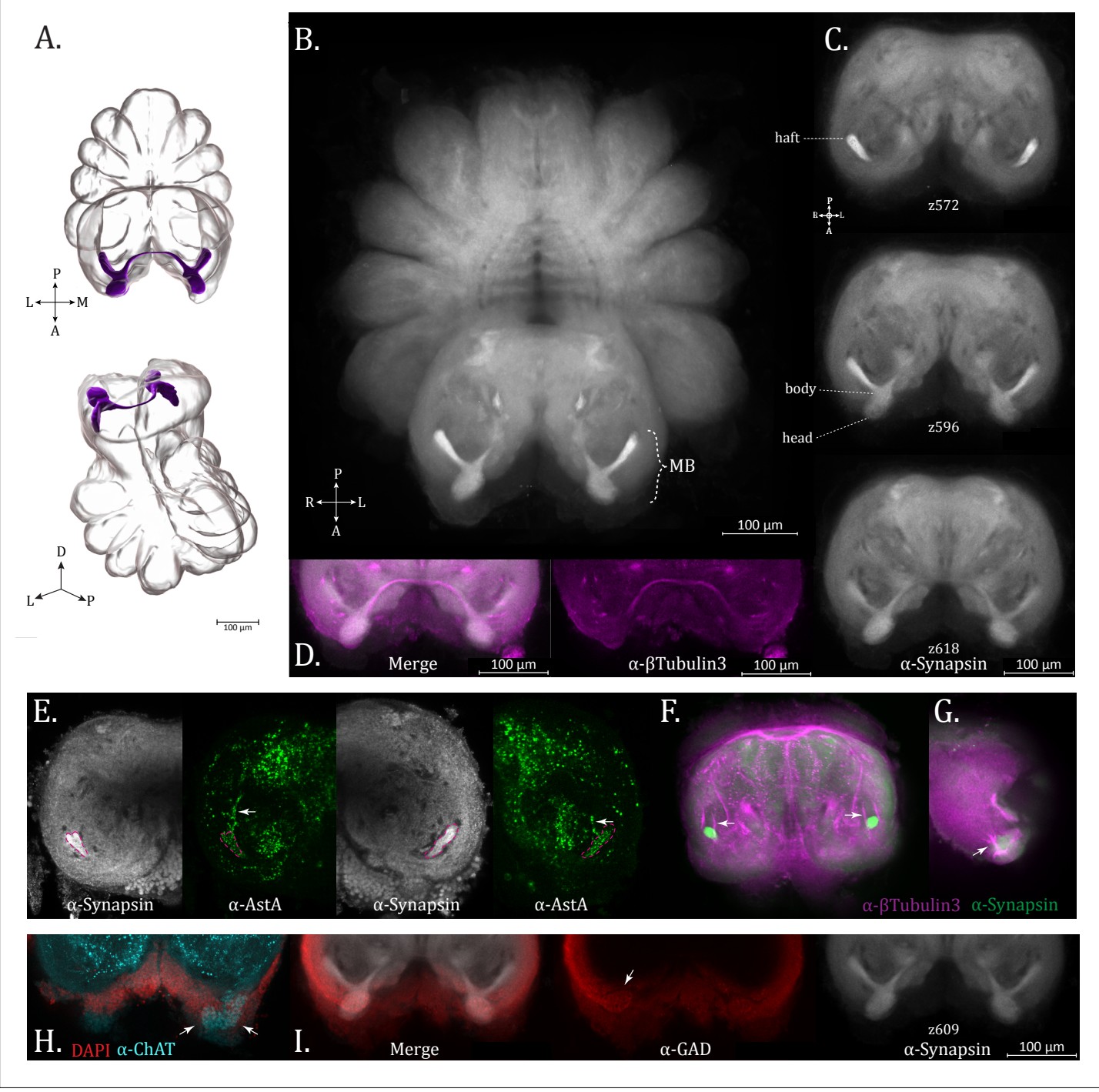

**Figure 7.** Mushroom bodies (MBs). (**A**) 3D rendering of MB neuropil as annotated from averaged α-synapsin volume, dorsal (top), and oblique posterior (bottom). (**B**) Maximum intensity projection of averaged α-synapsin volume, showing the mushroom bodies to be the most strongly immunoreactive structure in the supraesophageal ganglion. (**C**) Optical sections of the supraesophageal ganglion from an averaged α-synapsin volume (ventral (top) to dorsal (bottom)). The haft, body, and head regions of the MB are labeled. (**D**) α-βTubulin3 (magenta) immunoreactivity aligned with α-Synapsin volume (gray) (compare to bottom portion of previous subfigure) showing the arching form of the mid-line spanning MB bridge. (**E**) Allatostatin A immunoreactivity (α-AstA, green) present in the MB haft (pink dotted line marking location of α-synapsin immunoreactivity) with arrows pointing to innervation from the posterior side. (**F**) α-βTubulin3 (magenta) and α-Synapsin (green) immunoreactivity in the supraesophageal ganglion at the plane where the MB hafts appear (round, intensely immunoreactive). Arrows mark a fiber tract flanking the haft which could be the origin of the innervation in the preceding subfigure. (**G**) Tripart tract entering at the MB head to fuse with the tract descending through the MB. (**H**) Cropped optical section from an individual cholinergic stain (α-ChAT, cyan), of a plane just dorsal to the MB heads, showing putative globuli cells (arrows) within the protrusion of the

*Figure 7 continued on next page*

*Figure 7 continued*

secondary visual pathway, DAPI stain (red). (**I**) Cropped optical slice from standard brain (z609) at the level of MB heads, with Synapsin (α-synapsin, gray), GABAergic (α-GAD, red), and merged immunoreactivity, with arrow indicating potential innervation of the globuli cells. Compass abbreviations: A = anterior, P = posterior, D = dorsal, V = ventral, L = left, R = right.

The online version of this article includes the following video for figure 7:

**Figure 7—video 1.** Mushroom body 3D volume.

https://elifesciences.org/articles/107732/figures#fig7video1

The antero-dorsal input to the MB heads, representing the secondary eye pathway (*Strausfeld and Barth, 1993*), is much more conspicuous and has received considerable treatment within the literature. The MB heads are sometimes referred to as the third-order visual neuropil in this pathway, with the ample parallel fibers which give this structure its shape arising from globuli cells which cap the MB head.

The globuli cells are not distinguishable from the surrounding nuclei by DAPI signal, but can potentially be discerned through specific neurosignaling molecule immunostains. We find the cluster of cells closely associated with the MB heads is revealed by ChAT immunoreactivity, and to a lesser extent by GAD immunoreactivity, suggesting they represent cholinergic and GABAergic populations, respectively (*Figure 7H, I* – arrow(s)). Globuli cells in *C. salei* have previously been shown to be ChAT+ (*Fabian-Fine et al., 2017*). By βTubulin3 staining, we also observed a trident of tracts feeding into the dorsal aspect of the MB head (*Figure 7G*).

## Visual system

*U. diversus*, like many orb weavers, builds its web at night and can do so in essentially complete darkness in laboratory conditions, suggesting that vision is expendable to much of the spider's behavioral repertoire (*Eberhard, 1971*). Web-building spiders are considered to have poorer vision than spiders which depend on sight to capture prey, which is reflected in their diminished optic neuropils and tract pathways (*Long, 2021*; *Rivera-Quiroz and Miller, 2022*; *Steinhoff et al., 2024*).

The number and particularly size and spatial arrangement of spider eyes is variable and characteristic across species, but most commonly, spiders have eight eyes which give rise to two visual system pathways, the principal and secondary eye pathways (*Strausfeld and Barth, 1993*; *Strausfeld and Barth, 1993*; *Long, 2021*; *Steinhoff et al., 2024*). The pair of anterior medial eyes is also known as the principal eyes which innervate the principal pathway terminating primarily in the arcuate body. The remaining pairs are the six secondary eyes, including the anterior lateral eyes, posterior medial eyes, and posterior lateral eyes, whose projections form the secondary eye pathway. *U. diversus* has eight eyes which are approximately equal in size.

Relative to cursorial species (*Babu and Barth, 1984*; *Steinhoff et al., 2017*; *Steinhoff et al., 2024*), in *U. diversus* the anterior extensions of the protocerebrum containing the first and second-order optic neuropils are considerably thinner and not as extensively fused with the continuous neuropil of the supraesophageal mass, thus being prone to separating during dissection. Consequently, neither the primary nor secondary visual pathway neuropils appear reliably enough in the anti-synapsin volumes to be apparent in the averaged standard brain representation, but nevertheless, these structures are exhibited in various individual preparations. The optic neuropils in *U. diversus* tended to show weaker synapsin immunoreactivity but were clearly seen with antisera to HRP (*Figure 8A*).

As in other species, the secondary pathway is larger (*Figure 8B*), lifting away anteriodorsally from the zone of the MB heads. This continuity can be seen from the sliced three-dimensional maximum intensity projection of synapsin (*Figure 8B* – brace). The primary pathway is diminutive in *U. diversus* and emerges as a bulbous shape at the dorsal-most end of the brain through a field of somata (*Figure 8A*).

Previous reports have used GABA (*Becherer and Schmid, 1999*), histamine (*Schmid and Duncker, 1993*), dopamine (*Auletta et al., 2020*), CCAP (*Loesel et al., 2011*), and FMRFamide (*Becherer and Schmid, 1999*) to reveal the successive neuropils of the visual pathways. As noted above, the only features within the optic pathway for which we observed neurosignaling molecule immunoreactivity were the globuli cells with GAD and ChAT staining. It is possible that targets for which we could not acquire an effective antisera, such as histamine, could be revelatory of the optic lamellae and other visual pathway structures, as they have been for *C. salei* (*Schmid and Becherer, 1999*; *Schmid and*

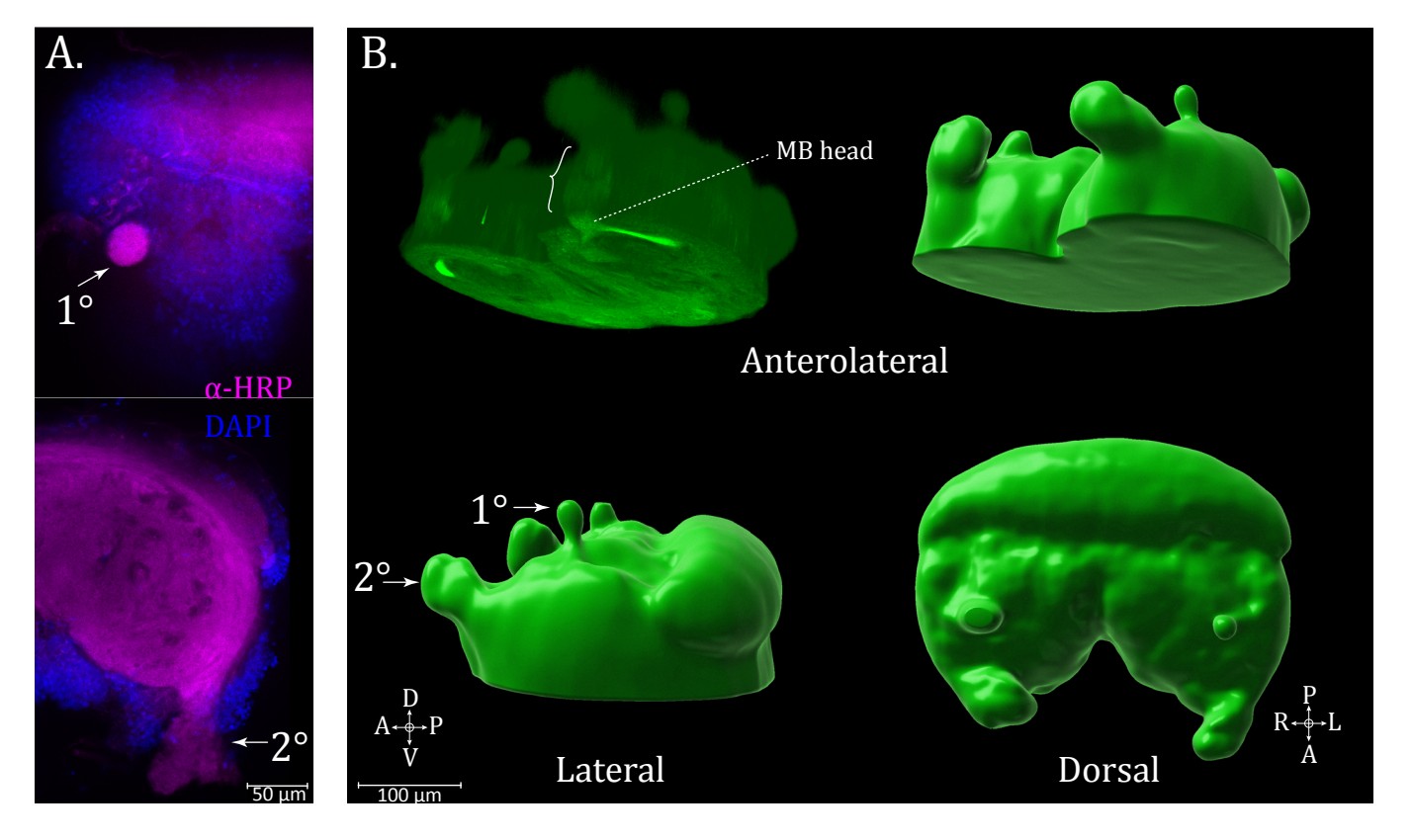

**Figure 8.** Visual pathways. (**A**) Immunostaining for α-HRP (magenta) for neuropil and use of DAPI (blue) for nuclei, arrows show the primary (1°) and secondary (2°) visual pathway extensions from the bulk of the supraesophageal tissue. (**B**) 3D renderings of synapsin immunoreactivity in the dorsal supraesophageal ganglion, with tissue of the primary (1°) and secondary (2°) visual pathway visible. Brace shows the path of the secondary pathway input to the mushroom body head. Compass abbreviations: A = anterior, P = posterior, D = dorsal, V = ventral, L = left, R = right.

*Duncker, 1993*). Specific compartments of the pathways, such as the medulla or lamellae, could not be confidently discerned with any preparation.

## Tonsillar neuropil

Within the historically non-descript central protocerebrum, we observed a synaptically dense neuropil structure in *U. diversus*. Beginning in the planes dorsal to the mushroom bodies, this paired structure is positioned directly on either side of the midline and is centrally located. Each half has an approximately ovoid appearance, particularly at the anterio-dorsal end, while being bridged at the posterior aspect. Between the two halves, at the midline, is a furrow which is negative for synapsin immunoreactivity, giving this neuropil, in conjunction with the synapsin-negative zone, a likeness to tonsils when viewed from the horizontal optical planes (*Figure 9A, C*– α-synapsin, *Figure 9—videos 1–3*) – hence our referral to this structure as the tonsillar neuropil.

   In individual anti-synapsin stains, a fiber tract traveling laterally adjoins this neuropil in the more dorsal–posterior portions. By tubulin immunoreactivity, it appears to bifurcate the structure below the bridge in the dorsal portion (*Figure 9B*). A subset of antisera for specific neuronal populations is instrumental in confirming this neuropil, as their immunoreactivity is circumscribed by its boundaries, with little neighboring signal to obscure the distinction (*Figure 9C*). Most representative among these is serotonergic immunoreactivity, exhibiting fine varicosities which neatly fill the area (*Figure 9Ci, vi*). TDC2+ signal is also prominent in this neuropil *Figure 9Cii, vii* – the relatively heavier terminals appear stronger on the periphery, and when viewed in alignment with the 5-HT channel, resemble a division of compartments, most notably in the ovoid regions where serotonin is found in an internal, core pattern, with octopaminergic/tyraminergic signal as a shell.

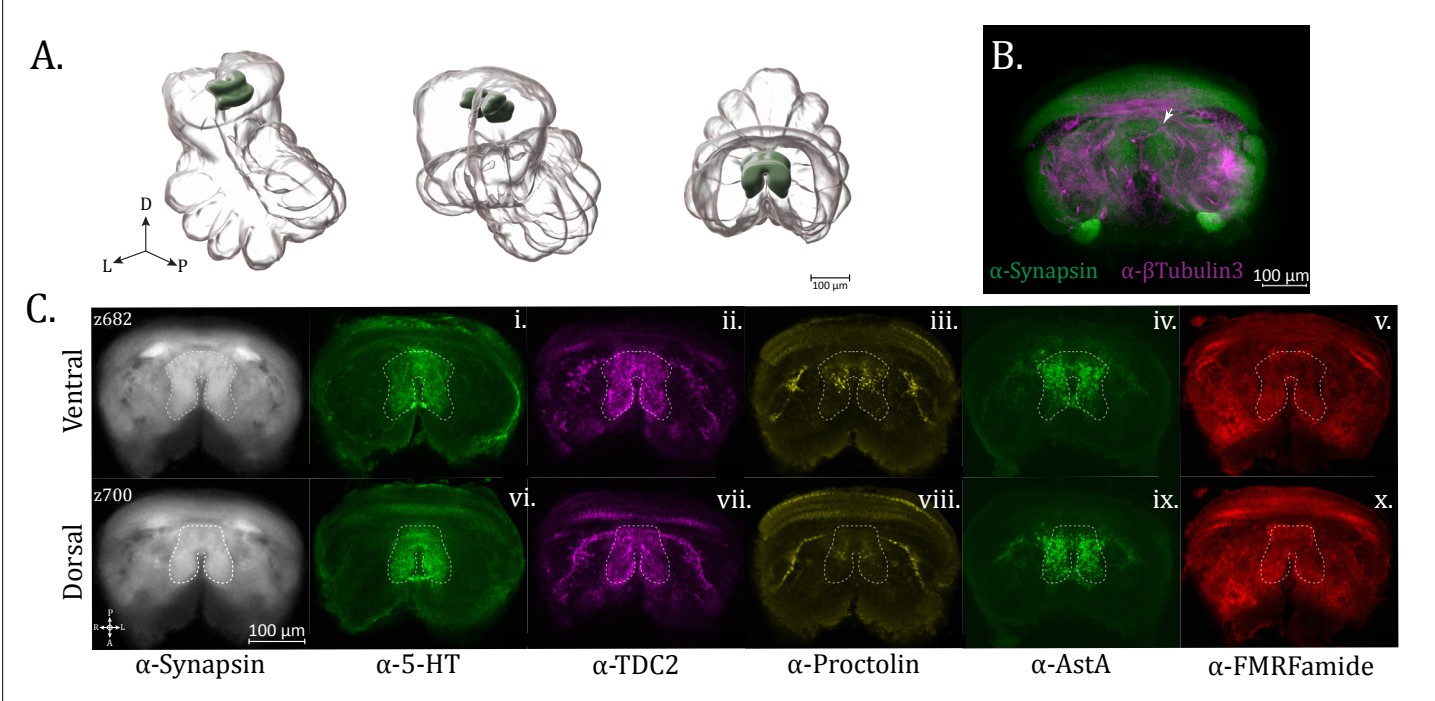

**Figure 9.** Tonsillar neuropil. (**A**) 3D rendering of tonsillar neuropil as annotated from averaged synapsin volume with posterior oblique, anterior oblique, and dorsal views, left to right. (**B**) Oblique horizontal optical section of supraesophageal ganglion with α-Synapsin (green) and α-βTubulin3 (magenta) immunoreactivity. The tonsillar neuropil is seen centrally, with the arrow denoting a fiber tract which passes medially across it. (**C**) Ventral (z682) and dorsal (z700) views of the tonsillar neuropil, demarcated by dashed lines, from the standard brain, showing expression of synapsin (gray), (i, vi) serotonergic (α-5-HT, green), (ii, vii) octopaminergic/tyraminergic (α-TDC2, magenta), (iii, viii) proctolin (α-Proctolin, yellow), (iv, ix) allatostatin A (α-AstA, green) and (v, **x**) FMRFamide (α-FMRFamide, red) immunoreactivity. Compass abbreviations: A = anterior, P = posterior, D = dorsal, V = ventral, L = left, R = right.

The online version of this article includes the following video(s) for figure 9:

**Figure 9—video 1.** Tonsillar neuropil 3D volume.
https://elifesciences.org/articles/107732/figures#fig9video1

**Figure 9—video 2.** Neurotransmitter immunostains in the tonsillar neuropil.
https://elifesciences.org/articles/107732/figures#fig9video2

**Figure 9—video 3.** Neuropeptide immunostains in the tonsillar neuropil.
https://elifesciences.org/articles/107732/figures#fig9video3

There may also be a division in the anterior–posterior dimension as allatostatin A immunoreactivity is more pronounced in the posterior bridging region and sparsely punctuated in the anterior ovoid zones (*Figure 9Civ, ix* – α-AstA), while proctolin immunoreactivity is limited to the posterior region (*Figure 9Ciii, viii* – α-proctolin). FMRFamide immunostaining is diffusely present, particularly in the anterior–dorsal portions of the tonsillar neuropil (*Figure 9Cv, x*), but this is amidst broadly saturated signal from this antibody throughout the protocerebrum.

## Protocerebral bridge

Originating antero-laterally and progressing posterior-medially through the ascending dorsal planes of the supraesophageal ganglion is a banded neuropil structure which we will designate as the protocerebral bridge (PCB). Wider in the lateral aspect, the structure tapers toward the medial end with the thinnest, midline-crossing component only being apparent in specific neuronal subpopulation stains. This is the dorsal-most neuropil seen in the interior of the protocerebrum before reaching the dense cap of somata (*Figure 10*; *Figure 10—videos 1–3*).

As with the previous neuropils, only a subset of antisera shows immunoreactivity within this structure. The most filling is GABAergic immunoreactivity (by anti-GAD stain) which defines a nearly complete swath of the neuropil with dense signal (*Figure 10Bi, iv* – α-GAD). Comparing further

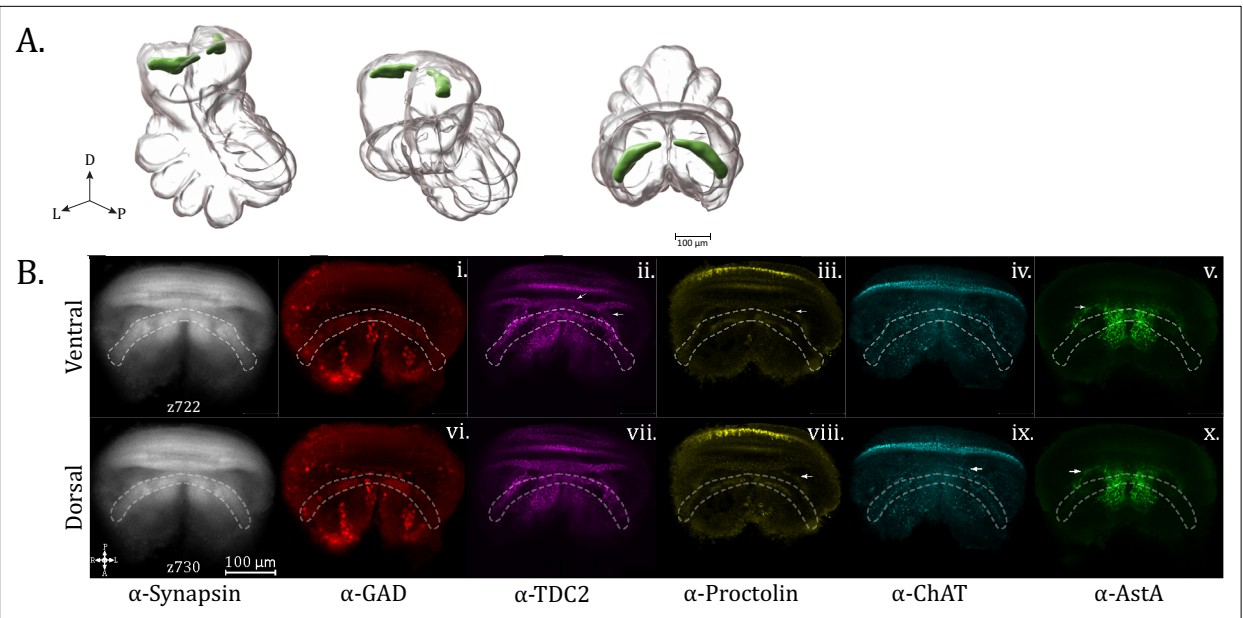

**Figure 10.** Protocerebral bridge (PCB) neuropil. (**A**) 3D rendering of PCB neuropil as annotated from averaged synapsin volume, with posterior oblique, anterior oblique, and dorsal views, left to right. (**B**) Ventral (z722) and dorsal (z730) views of the PCB, as demarcated by dashed lines, from the standard brain, showing expression of synapsin (gray), (i, vi) GABAergic (α-GAD, red), (ii, vii) octopaminergic/tyraminergic (α-TDC2, magenta), (iii, viii) proctolin (α-Proctolin, yellow), (iv, ix) cholinergic (α-ChAT, cyan), and (v, x) allatostatin A (α-AstA, green) immunoreactivity. Arrows point to the dorsal protocerebral commissure (dPCC). Compass abbreviations: A = anterior, P = posterior, D = dorsal, V = ventral, L = left, R = right.

The online version of this article includes the following video(s) for figure 10:

**Figure 10—video 1.** Protocerebral bridge 3D volume.
https://elifesciences.org/articles/107732/figures#fig10video1

**Figure 10—video 2.** Neurotransmitter immunostains in the protocerebral bridge.
https://elifesciences.org/articles/107732/figures#fig10video2

**Figure 10—video 3.** Neuropeptide immunostains in the protocerebral bridge.
https://elifesciences.org/articles/107732/figures#fig10video3

neuronal subtype preparations reveals that the PCB has a layered structure. TDC2 immunoreactivity is pronounced throughout the length of the protocerebral bridge, displaying heavy chains of puncta on the posterior edge of the bridge (*Figure 10Bii, vii* – α-TDC2). Proctolin immunoreactivity forms a tight, thinner band, primarily at the medial end of the neuropil, comprised of fine puncta and is centrally located among the layers (*Figure 10Biii, viii* – α-proctolin). The cholinergic pattern is a distinct thin layer on the anterior and posterior edge, most clearly visible on the lateral portion of the PCB (*Figure 10Biv, ix* – α-ChAT). A faint section of AstA immunoreactivity was also seen (*Figure 10Bv, x* – α-AstA).

Posterior to the PCB is an arching string of varicosities which reaches its apex just before the appearance of the dorsal arcuate body layer. We refer to this as the dorsal protocerebral commissure (*Figure 10B* – arrows) and the neuronal subtype populations which show PCB expression tend to also innervate this commissure. The strongest of these are octopaminergic/tyraminergic neurons (anti-TDC2) and proctolin+ neurons (anti-proctolin) (*Figure 10B* – α-TDC2, α-proctolin). This pathway separates from the posterior–lateral contour of the PCB, where there is an approximately triangular expansion of immunoreactivity before traveling medially and passing just posterior to the edge of the tonsillar neuropil. A similar pattern is also seen for allatostatin A (*Figure 10B* – α-AstA). Cholinergic immunoreactivity is also apparent in the posterior arch, but is more subtle and in a single layer in the anterior domain, less comparable to that of TDC2, proctolin, and allatostatin A.

## Arcuate body

The arcuate body (AB) is a prominent neuropil structure found in all spider species whose central nervous system anatomy has been examined closely to date. Residing in the dorso-posterior aspect

of the supraesophageal mass, this solitary crescent-shaped structure has been recognized as having at least two broad divisions, the ventral and dorsal lobes (*Hanström, 1928*; *Loesel et al., 2011*; *Steinhoff et al., 2017*).

In *U. diversus*, at the grossest level, we likewise observe two lobes of the arcuate body, which we will refer to as the ventral (ABv) and dorsal (ABd) lobe (*Figure 11A, B*; *Figure 11—videos 1–3*). Though largely coincident in the dorso-ventral axis, the ventral arcuate body somewhat envelopes the dorsal arcuate lobe, hence appearing first from the ventral direction and lingering posteriorly on the dorsal side, with only a smaller part of the dorsal arcuate body protruding independently beyond the ABv at the dorsal end (*Figure 11B*; *Figure 11—video 1*).

Each lobe (ABv and ABd) can be further subdivided into two sub-lobes or layers – a posterior (posABv and posABd) and anterior (antABv and antABd) section, making a total of four units (*Figure 11C*). The sublayers of the arcuate body lobes are distinguishable by immunostaining for specific neuronal subpopulations that differentially innervate the layers (*Figures 11D and 12*). By examining these expression patterns, another tier of complexity can be appreciated, as each of these sublayers (posABv, antAbv, posABd, and antABd) can be further subdivided into two or even three aspects, depending on the antisera used.

Posterior to the arcuate body is a crest of somata which has been previously referred to as the posterior cell layer (PCL) (*Fabian-Fine et al., 2017*). Neurons of the PCL send their projections anteriorly through the ventral arcuate layers, as revealed by immunostaining for βTubulin3 in conjunction with synapsin (*Figure 11E*). The fibers successively appear more medially as one progresses further dorsally in the arcuate lobes, with certain tracts being thicker than others. Hill noted the presence of tracts running through the arcuate body to join the PCDt in the jumping spider, *P. johnsoni* (*Hill, 1975*).

The innervation pattern of a given neuronal subpopulation in a layer of the arcuate body is not a general delineation of the structure of that layer, as different transmitter populations can display distinct expression patterns within the same layer. An example is the dorsal arcuate body (ABd), where TDC2 immunoreactivity shows a prominent columnar, flagstone innervation, while proctolin has a sparse field of fine puncta in the same layer (*Figure 11F*). Even though the arcuate body is organized in lateral bands, columnar organization can also be observed in several layers. For example, a series of discernible units of allatostatin A+ immunoreactivity are seen in the posABv layer, suggesting the columnar organization which is present, but generally obscured by the density of staining (*Figure 11G*).

There is a diversity of layering patterns (*Figure 12*), but some basic motifs emerge. Innervation can be partial, as in filling a single sublayer (anterior or posterior) of a lobe, or complete throughout the lobe, taking on a saturated appearance, a meshwork of neurites, or a sparse field of puncta. The space between marked sublayers may have finer neurite connections which have been described as palisade-like (*Loesel et al., 2011*). At the dorsal end of the ventral arcuate body (ABv), heavy garland-like varicosities may form, in certain examples (α-proctolin, α-ChAT, *Figure 12* – arrowheads, on respective subpanels) appearing as disjointed units, suggestive of an undergirding column. More prevalent in the dorsal arcuate, a robust networking of thicker immunoreactive fibers weaves between roughly trapezoidal signal-negative areas (α-5-HT, α-TDC2, *Figure 12* – braces, on respective subpanels), resembling a flagstone pathway. Detailed descriptions of arcuate body layer projection patterns (*Figure 12*) and comparisons to other spider species are found below in the respective subsections for each neurosignaling molecule.

## GABA

In the ventral lobe of the arcuate body, GAD immunoreactivity is faint and difficult to discern from bleed-through or background; however, we see a clear illustration of how neurites stemming from the somata of the PCL extend through the arcuate body layers (*Figure 12i* – α-GAD, ventral – arrows). At the edge of the posterior ABd is a layer of diffuse immunoreactivity (*Figure 12ii* – α-GAD, dorsal – arrow). Moving anteriorly, this is followed by a thin layer of puncta (*Figure 12ii* – α-GAD, dorsal – arrowhead), which may be connected through minute projections to the next layer which is thicker than the first (*Figure 12ii* – α-GAD, dorsal – brace). The pattern and relative proportions of the layers are broadly similar in *C. salei* (*Becherer and Schmid, 1999*; *Fabian-Fine et al., 2015*), where the anterior-most layer in the dorsal arcuate body is also the thickest, albeit less densely stained in our example.

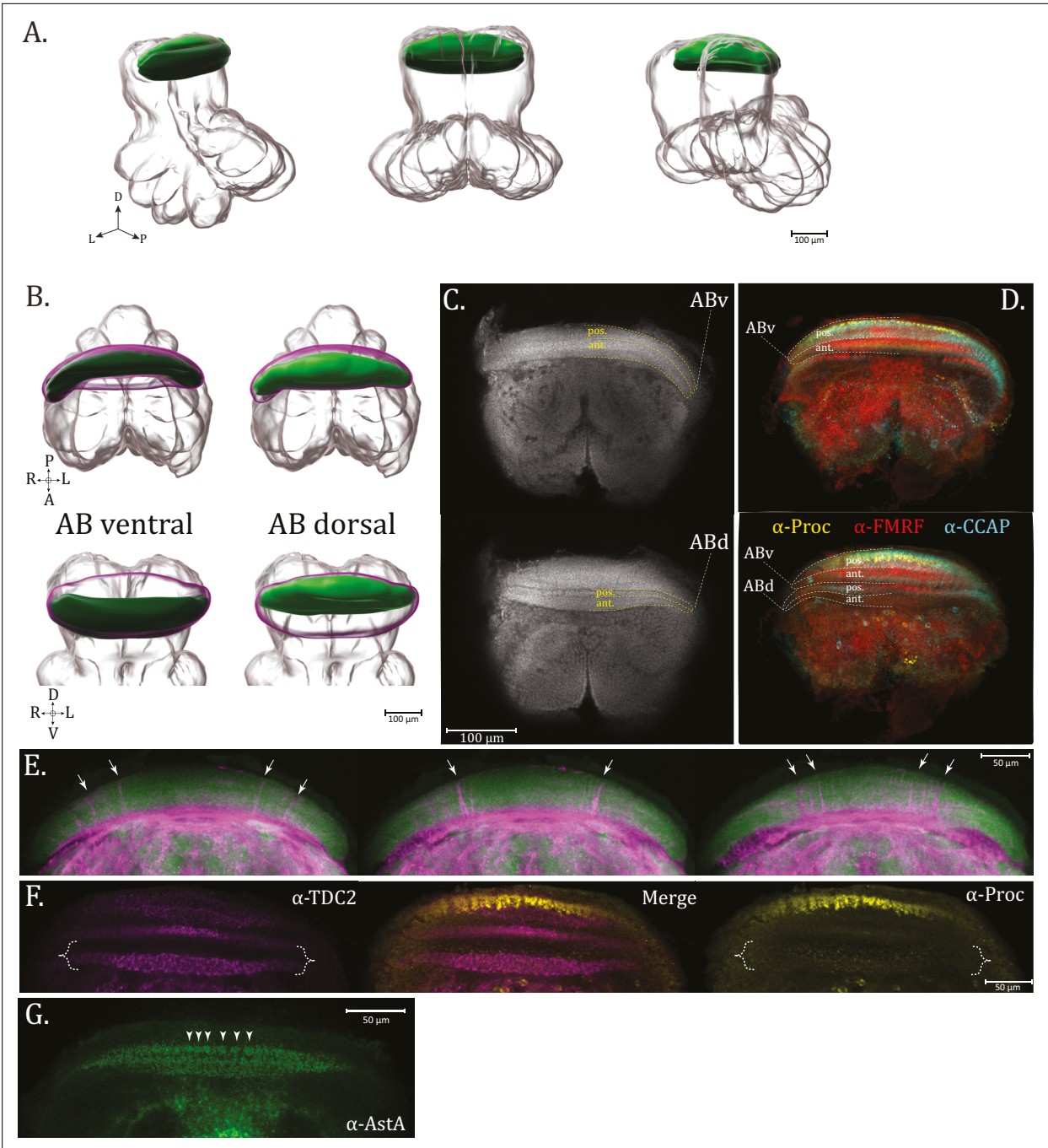

**Figure 11.** Arcuate body. (**A**) 3D rendering of arcuate body neuropil as annotated from averaged α-synapsin volume, posterior oblique, posterior, and anterior oblique views, left to right, respectively (**B**) Individual 3D rendering of the ventral arcuate body lobe (ABv, dark green) and dorsal arcuate body lobe (ABd, light green), with magenta envelope representing space which would be occupied by the missing lobe in each image. Top row images are dorsal views, bottom row images are oblique posterior (**C**). Optical horizontal slices of α-synapsin immunoreactivity from the dorsal supraesophageal ganglion. The top image is ventral relative to the bottom and shows the ventral arcuate body lobe (ABv), while the bottom image features both the ventral and dorsal arcuate body lobe (ABd). Each lobe contains an anterior (ant.) and posterior (pos.) section, marked with yellow dashed lines. (**D**) Ventral (top) and dorsal (bottom) views showing aligned image volumes of proctolin (α-Proctolin, yellow), crustacean cardioactive peptide (α-CCAP, cyan), and FMRFamide (α-FMRFamide, red) immunoreactivity, demonstrating distinct structures as well as overlapping innervation of the arcuate body layers. (**E**) α-βTubulin3 (magenta) and α-Synapsin (green) immunoreactivity in the arcuate body (ventral to dorsal as top to bottom, respectively), with arrows marking where pronounced fiber tracts pass through the arcuate body layers. (**F**) Dorsal view of arcuate body showing layering in ABv and ABd (brace), for proctolin (α-Proctolin, yellow) and octopaminergic/tyraminergic (α-TDC2, magenta) immunoreactivity which have overlapping but

*Figure 11 continued on next page*

*Figure 11 continued*

distinct innervation patterns in the anterior ABd. (**G**) Arrowheads marking patterning of individual units of α-AstA immunoreactivity in the arcuate body, suggestive of a columnar structure. Compass abbreviations: A = anterior, P = posterior, D = dorsal, V = ventral, L = left, R = right.

The online version of this article includes the following video(s) for figure 11:

**Figure 11—video 1.** Arcuate body 3D volume.
https://elifesciences.org/articles/107732/figures#fig11video1

**Figure 11—video 2.** Neurotransmitter Immunostains in the arcuate body.
https://elifesciences.org/articles/107732/figures#fig11video2

**Figure 11—video 3.** Neuropeptide immunostains in the arcuate body.
https://elifesciences.org/articles/107732/figures#fig11video3

## Acetylcholine

In the arcuate body, cholinergic signal is predominantly found in the ventral arcuate body lobe (*Figure 12iii* – α-ChAT, ventral). Within the ventral side of this lobe, cholinergic immunoreactivity forms fine puncta which completely fill the anterior sub-layer of this lobe (*Figure 12iii* – α-ChAT, ventral – brace). Toward the dorsal end of this lobe, the punctate immunoreactivity forms heavier beaded varicosities (*Figure 12iv* – α-ChAT, dorsal – arrowhead). Midway, there are faint column-like expression patterns joining from a thin layer within the posterior ventral AB (pABv). A broader layer is innervated on the anterior side of the dorsal lobe (*Figure 12iv* – α-ChAT, dorsal – brace), a sparser example of the flagstone-type patterning seen more clearly with α-TDC2 immunoreactivity, below. Acetylcholinesterase activity has been visualized through acetylthiocholine iodide staining in cursorial and web-building species (*Meyer and Idel, 1977*; *Meyer and Pospiech, 1977*). The lycosid *T. spinipalpis* and salticid *M. muscosa* appear to show at least two broad layers of cholinergic signal, similar to *U. diversus* (*Meyer and Idel, 1977*). The orb-web-building araneid *A. sclopetarius* had relatively diminished arcuate body signal compared to the central protocerebrum of the cursorial species with this preparation (*Meyer and Pospiech, 1977*), and the multilayered signal was not as clear. Since it can be difficult to judge which arcuate body lobes are on display from a single slice, it is also possible that the diminished signal in the anterior layer for *A. sclopetarius* might be consistent with *U. diversus* patterning, if that is indeed the dorsal arcuate body lobe.

## Dopamine

Within the arcuate body, TH immunoreactivity occupies a single layer in the posterior aspect of the dorsal lobe (*Figure 12vi* – α-TH, dorsal – arrow), supplied by thin and sparse neurites stretching from the anteriorly located tracts. This single layer of punctate terminals with anteriorly branching projections is consistent with both *H. lenta* and *P. regius* (*Auletta et al., 2020*), but otherwise, the *U. diversus* dopaminergic arcuate body layering appears simpler and more comparable to the jumping spider, *P. regius*, due to lacking the additional wispy immunoreactivity in anterior layers as in *H. lenta*.

For the wolf spider, *H. lenta*, TH labeling reveals densely stained first and second-order optic neuropils (*Auletta et al., 2020*). In contrast, we see a stark lack of immunoreactivity in anterior regions which would be expected to contain the comparable neuropils in *U. diversus*.

## Serotonin

Immunostaining against 5-HT in the social huntsman, *Delana cancerides,* shows two gross levels of immunoreactivity in the arcuate body; a wide diffuse layer of puncta, and a thinner layer bordered by dense puncta on each side, with columnar-like expression in between (*Strausfeld et al., 2006*). Taken together as two adjacent layers, this pattern is remarkably similar to that seen for our model species. In *U. diversus*, serotonergic immunoreactivity shows a faint layer in the posterior ventral arcuate lobe (*Figure 12vii* – α-5-HT, ventral – arrow), and an anterior ventral arcuate sublayer broadly flush with minutely fine fibers (*Figure 12vii* – α-5-HT, ventral – brace). The dorsal arcuate lobe displays a robust and wide immunoreactive pattern resembling the flagstone pattern, hinting at a columnar structure (*Figure 12viii* – α-5-HT, dorsal – brace). Innervation in this layer greatly resembles that seen for TDC2 in the same lobe (*Figure 12* – α-TDC2, dorsal – brace).

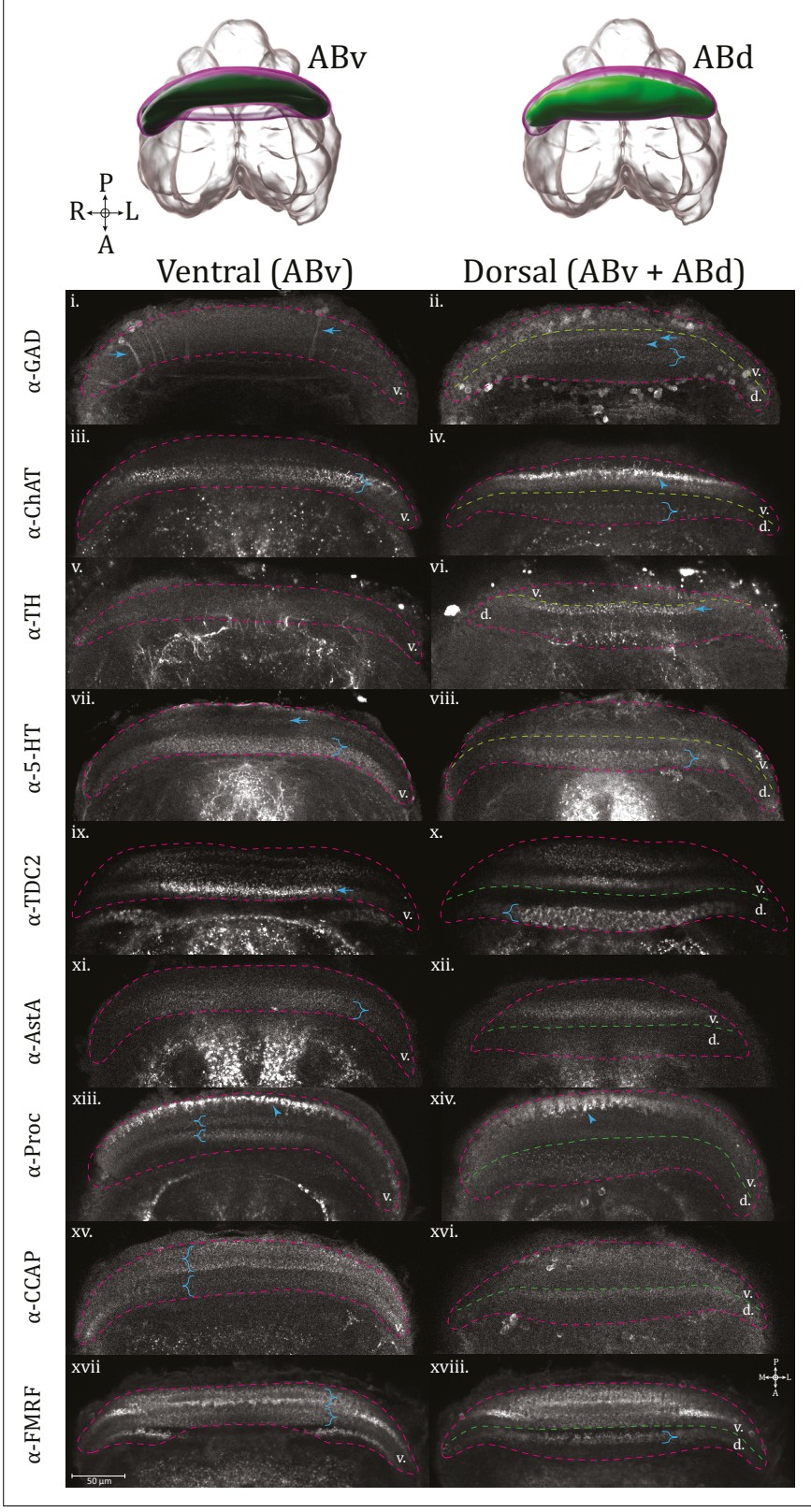

**Figure 12.** Arcuate body layers revealed by staining for specific neurosignaling populations. Ventral (left column, odd roman numerals) and dorsal (right column, even roman numerals) horizontal optical section views of the arcuate body lobes: ventral AB, represented in the 3D rendering (dark green), and dorsal AB (light green). The perimeter of the whole arcuate body (ABv and ABd) is outlined (dashed line, magenta), as well as the boundary

*Figure 12 continued on next page*

*Figure 12 continued*

(dashed line, green) between ABv (labeled internally as 'v') and ABd (labeled internally as 'd'). Immunoreactivity for the following targets is shown: **GABAergic** (α-GAD) (i, ii), arrows in the ventral image marking neurites traversing arcuate body layers from the posterior cell layer, arrow and arrowhead in the dorsal image mark posterior ABd innervation patterns, brace in the dorsal image marks a thicker innervation pattern in the anterior ABd sublayer. **Cholinergic** (α-ChAT) (iii, iv), in the ventral image, the brace marks a fine puncta layer in the anterior ABv sublayer, in the dorsal image, an arrowhead marks pronounced varicosities in the dorsal aspect of the anterior ABv, while a brace marks the flagstone-pattern innervation seen in the anterior sublayer of the ABd. **Dopaminergic** (α-TH) (v, vi), in the dorsal image, an arrow marks a layer of innervation in the posterior ABd. **Serotonergic** (α-5-HT) (vii, viii), in the ventral image an arrow marks a faint layer of immunoreactivity in the posterior ABv, and a brace marks broad, fine innervation in the anterior ABv. In the dorsal image, a brace marks the width of the anterior ABd layer showing flagstone-pattern innervation. **Octopaminergic/tyraminergic** (α-TDC2) (ix, x), in the ventral image, an arrow marks dense varicosities in the anterior ABd sublayer. In the ventral image, a brace marks pronounced flagstone-pattern innervation in the anterior ABv sublayer. **Allatostatin A** (α-AstA) (xi, xii), in the ventral image a brace marks the innervation of the anterior ABv. **Proctolin** (α-Proctolin) (xiii, xiv), in the ventral image an arrowhead marks intense immunoreactive terminals at the posterior ABv, and the stacked braces define the boundaries of innervation to the posterior (top) and anterior (bottom) sublayers of the anterior ABv. In the dorsal image an arrowhead marks an example of garland-like varicosities visible at the dorsal end of the ABv. **Crustacean cardioactive peptide** (α-CCAP) (xv, xvi), in the ventral image stacked braces mark the posterior (top brace) and anterior (bottom brace) layers of the ABv. **FMRFamide** (α-FMRFamide) (xvii, xviii), in the ventral image stacked braces mark the boundaries of posterior (top) and anterior (bottom) layers of the ABv. In the dorsal image the brace marks innervation to the posterior ABd layer. Compass abbreviations: A = anterior, P = posterior, D = dorsal, V = ventral, L = left, R = right.

## Octopamine/tyramine

Both anterior and posterior sublayers of the ventral arcuate body exhibit TDC2 immunoreactivity (*Figure 12ix* – α-TDC2, ventral). The posterior ABv is saturated with diffuse puncta, which are also seen anteriorly, along with denser varicosities in this sublayer (*Figure 12ix* – α-TDC2, ventral – arrow). In the dorsal arcuate body lobe, TDC2 immunoreactivity appears prominently in the anterior sublayer (aABd), where it fully fills the span of this layer with robust staining resembling a series of flagstone-shaped columnar-like elements (*Figure 12x* – α-TDC2, dorsal – brace). Octopaminergic expression has been reported in the arcuate body (labeled 'central body' in source) (*Seyfarth et al., 1993*) of *C. salei*, where a parasagittal section shows strong immunoreactivity in the ventral portion of both arcuate body lobes. We must imagine the respective horizontal view, but it would appear by the gaps in immunoreactivity that a dorsal horizontal slice in *C. salei* should show three general layers of AB staining, which is essentially what we see from a dorsal plane in *U. diversus*.

## Allatostatin A

The pattern of arcuate body innervation by AstA+ neurons is in general agreement with findings from *C. salei* and *M. muscosa* (*Loesel et al., 2011*; *Steinhoff et al., 2017*) where signal is prominent in the ventral arcuate lobe (ABv), with little staining in the dorsal arcuate (ABd) (*Figure 12xi, xii* – α-AstA). Concerning the sublayers of the ventral arcuate lobe, AstA immunoreactivity is seen primarily on the anterior side of the posterior ventral arcuate (posABv) and fully encompasses the anterior ventral arcuate (antABv) (*Figure 12xi, xii* – α-AstA – brace).

## Proctolin

Proctolin immunoreactivity is evident in all lobes and layers of the arcuate body (*Figure 12* – α-proctolin). In the ventral arcuate body (ABv), at the posterior end, there is a line of intense terminals (*Figure 12xiii* – α-proctolin, ventral – arrowhead), with more anterior layers filled with diffuse puncta. Toward the dorsal end of the ventral arcuate body (ABv), the proctolin immunoreactivity in the posterior-most layer transforms into heavy garland-like columnar varicosities extending at an anterio-dorsal angle (*Figure 12xiv* – α-proctolin, dorsal – arrowhead). This is in complete correspondence to the varicosities seen for this layer in *C. salei* (*Loesel et al., 2011*). In the anterior ventral arcuate (antABv), the anterior and posterior sublayers (*Figure 12xiii* – α-proctolin, ventral – braces) take on an intricate mesh-like form, also with smaller flagstone formations. Between these two layers are fine palisade neurites. Both sublayers of the dorsal arcuate (ABd) are also filled, but with a sparse field of fine puncta (*Figure 12xiv* – α-proctolin, dorsal).

## CCAP

CCAP expression is strong in the posterior ventral AB layer with a fine mesh, punctate appearance which seemingly contours the columnar structures on the anterior and posterior boundaries of this layer. The anterior and posterior sublayers of the ventral AB are highlighted, with a decrease in staining within the area between the sublayers (*Figure 12xv* – α-CCAP, ventral – braces). In the anterior ventral AB, the anti-CCAP expression is slightly finer and more punctate than in the posterior layer. Within the dorsal AB (*Figure 12xvi* – α-CCAP, dorsal), only the posterior layer has appreciable expression, showing a single, finely innervated but moderately thick layer on the posterior boundary of the dorsal AB. CCAP-immunoreactive layers in *U. diversus* are comparable to *C. salei* (*Loesel et al., 2011*), as for both species the thickest staining layer is the most posterior one (ventral arcuate body lobe), followed anterio-dorsally by a lesser layer, and with a thinner strand of intensely immunoreactive boutons running through the more anteriorly located dorsal arcuate body lobe. CCAP immunoreactivity in two distinct layers of the arcuate body appears to be a consistent characteristic of the arcuate body lobes across arachnids, with the source also being repeatedly attributable to the 'l' and 'dl' cluster neurons (*Breidbach et al., 1995*). While we also see abundant CCAP+ somata, the limited expression within their projections does not allow us to confirm whether similar clusters are innervating the arcuate body layers in *U. diversus*.

## FMRFamide

From both *C. salei* (*Becherer and Schmid, 1999*) and the giant house spider, *Tegenaria atrica* (*Wegerhoff and Breidbach, 1995*), a basic structure of the FMRFamide-immunoreactive arcuate body layers emerges, where the entire dorsal arcuate body lobe is suffused with immunoreactivity, there is a sharp strand of garland-like varicosities giving way to the typical columnar arrangement in the posterior dorsal arcuate body layer (posABd), and more diffuse, punctate immunoreactivity in the anterior dorsal arcuate body layer (antABd). This pattern is approximately what we see in *U. diversus*, with additional details visible in the stacked image volume (*Figure 12xvii, xviii* – α-FMRFamide).

It appears that the saturated signal within the ventral arcuate lobe is actually the result of an innervation pattern which is stronger in the wall of each tubular-like sublayer, and weaker in the interior. This can be seen from several specific planes which slice longitudinally through both sublayers, revealing four layers, each being the boundary of one of the sublayers (*Figure 12xvii* – α-FMRFamide, ventral – braces). In the dorsal arcuate body, the immunoreactivity is primarily in the posterior sublayer, having the heavy varicosities at the ventral aspect (*Figure 12xviii* – α-FMRFamide, dorsal – brace). Relative to other examined spiders, the punctate pattern in the anterior sublayer is weakly present. The arcuate body layering pattern of FMRFamide immunoreactivity is similar to that of CCAP.

## Discussion

Almost the entirety of spider central nervous system literature has been studied from tissue slices, with few examples of whole mounts (*Auletta et al., 2020*; *Breidbach et al., 1995*; *Schmid and Duncker, 1993*; *Steinhoff et al., 2017*; *Steinhoff et al., 2024*). Our ability to observe novel structures and make comparisons between innervation patterns was aided by whole-mount preparation and averaged brain alignment, which allowed us to compare the relative expression of targets within the same reference volume. Furthermore, imaging and alignment of many neurosignaling molecule stains in a single species were clarifying for the identification of novel structures, as a subset of stains crystallized putative boundaries. While a dozen or so species have been studied for the expression patterning of individual neurosignaling molecules (*Auletta et al., 2020*; *Breidbach et al., 1995*; *Hwang et al., 2015*; *Moon and Tillinghast, 2013*; *Schmid and Becherer, 1999*; *Steinhoff et al., 2017*; *Strausfeld et al., 2006*), the wandering spider, *C. salei*, is essentially the only species prior to the current work to have been the subject of sustained efforts to amass such neuroanatomical descriptions for most of the canonical neurotransmitters and neuromodulators. Given the utility of specific stains for understanding of neuropil structures (*Figure 13*, *Figure 13—video 1*), tracts, and other features, this atlas provides a rich source for comparative anatomy in an orb-weaving spider, *U. diversus*, while also extending knowledge of a number of different neurosignaling pathways for spiders at large.

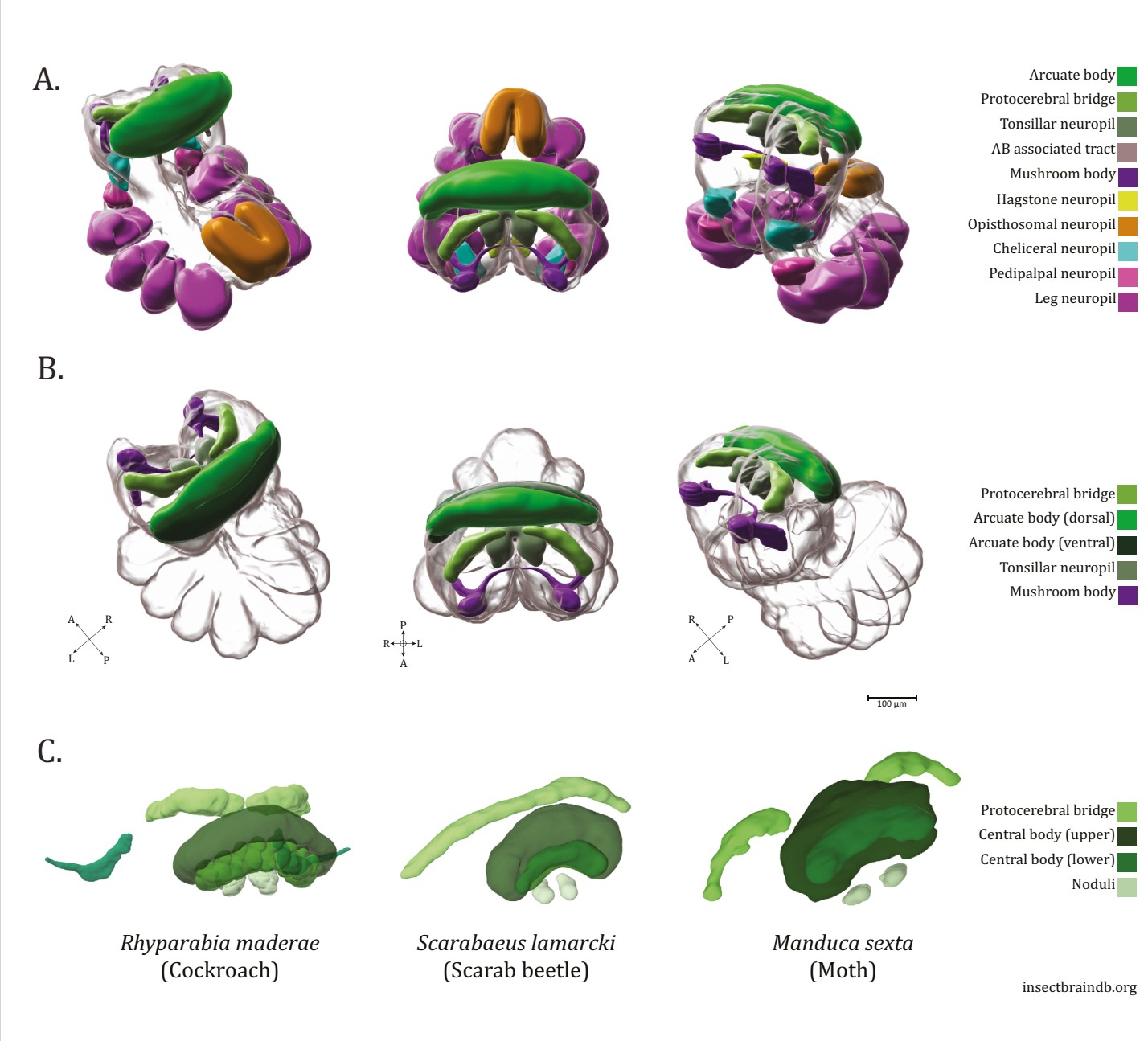

**Figure 13.** Summary of *U. diversus* neuropils. (**A**) Posterior oblique, dorsal, and anterior oblique views of 3D renderings of the standard *U. diversus* brain with annotations of major established and novel neuropils throughout the subesophageal and supraesophageal masses. (**B**) 3D renderings of the standard *U. diversus* brain with annotations of potential central complex constituents in shades of green (protocerebral bridge, arcuate body ventral and dorsal lobes, tonsillar neuropil), also showing the mushroom body (purple). (**C**) 3D neuropil renderings from neuropils of the central complex as found in the insects *Rhyparabia maderae*, *Scarabaeus lamarcki*, and *Manduca sexta* (images from https://insectbraindb.org/app/). Compass abbreviations: A = anterior, P = posterior, D = dorsal, V = ventral, L = left, R = right.

The online version of this article includes the following video for figure 13:

**Figure 13—video 1.** 3D rendering of all identified brain regions.

https://elifesciences.org/articles/107732/figures#fig13video1

## Subesophageal features

Explorations of neurosignaling population innervation in the subesophageal mass have generally been less detailed than within the supraesophageal. Certain neuropeptides were either only briefly shown to be immunoreactive (such as allatostatin A, *Steinhoff et al., 2017* or proctolin, *Groome et al., 1991*) or slices for those particular stains were not provided from the subesophageal ganglion, leaving it ambiguous as to whether immunoreactivity is present there (e.g. CCAP, *Loesel et al., 2011*). We find that all neuropeptidergic antisera, as well as the others examined in this study, have robust expression throughout the subesophageal mass. One observation that does not appear to be previously noted is that there is a roughly equal anterior/posterior division in the leg neuromeres. Whereas some immunostains reveal equal innervation of the halves (α-TH), others show divergent patterns (α-TDC2), or predominant expression in only one compartment (α-AstA). Based on select examples where the origin of innervation is discernible, the posterior and anterior compartments of the LNs may be supplied by neurites from different tracts within the interior of the subesophageal ganglion. The anatomical differences noted here may be due to different functional roles of these neurons in the LNs. For example, mechanosensory neurons in the leg receive input from octopaminergic and tyraminergic efferents, which increase the gain in mechanosensory touch responses (*Sukumar et al., 2018*; *Widmer et al., 2005*).

The OpN is a section of the subesophageal ganglion which has received relatively less attention. The preeminent reference for major tracts within the spider synganglion is the treatment in *C. salei* (*Babu and Barth, 1984*), but despite a detailed annotation throughout the synganglion, the trajectories within the opisthosomal ganglion were not diagrammed. A more recent expansion of this anatomical knowledge to further cursorial as well as web-based species of spiders (*Steinhoff et al., 2024*) likewise did not comment on the OpN. A depiction from *Hanström, 1928* shows that longitudinal tracts run parallel to the midline, as well as more laterally, and that there are crossing branches between them, forming a ladder-like architecture. This bears a resemblance to the pattern revealed by specific antisera in *U. diversus*, confirming the central tracts, perimeter defining tracts, as well as crossing fibers within the opisthosomal ganglion – though whether they cross completely from midline to periphery was not apparent. In certain preparations, we observed a ladder structure as well as a ring-like central structure with neurites projecting like spokes. Further studies revealing the sources and destinations of these tracts will benefit understanding this intermediary neuropil at the transition between prosoma and opisthosoma.

## Mushroom bodies

As evident from synapsin volumes, the mushroom bodies of *U. diversus* are the most salient feature in the central supraesophageal ganglion. The *U. diversus* MBs have a complete appearance, exhibiting an attached haft region similar to visually dependent spiders (*Steinhoff et al., 2024*), and to which we find evidence suggesting innervation, albeit from an unknown origin. Historically, the MBs have at times been referred to as the third-order visual neuropil and have been discussed in the context of the visual pathways, which form the subject of a substantial portion of the spider nervous system literature (*Hill, 1975*; *Rivera-Quiroz and Miller, 2022*; *Steinhoff et al., 2017*; *Steinhoff et al., 2024*; *Strausfeld et al., 1993*; *Strausfeld and Barth, 1993*). The optic neuropils of *U. diversus* are diminutive, which is consistent with hunting through mechanosensation on a web. While we employed several neurotransmitter stains which have identified upstream optic pathway elements (e.g. medulla, lamellae) in other species, these first and second-order structures were not evident even in preparations where the labile tissue of the secondary pathway was intact. The diminished nature of the optic pathways, but simultaneous presence of a distinct MB structure in *U.* diversus, raises an incongruence concerning the role of the MB. A growing literature is suggestive of a deeper complexity, as examples of both cursorial and web-based spiders can be found which either have or lack MBs (*Long, 2016*; *Long, 2021*; *Steinhoff et al., 2024*). The fact that such synaptically dense structures persist in spider species whose visual capacities seem all but irrelevant to their lifestyle indicates the sensory input to the mushroom bodies may differ between species. Mushroom body-like neuropils appear broadly among arthropods, suggesting a deep evolutionary origin despite variation in their form (*Wolff and Strausfeld, 2015*). The mushroom bodies of insects, as most granularly understood in *Drosophila melanogaster*, were originally considered to be olfactory integration centers, and while remaining the most apparent input, subsequent studies have shown this center to also process multiple sensory

modalities and influence behaviors not directly related to olfaction (*Aso et al., 2014*). Evolutionary pressures on certain species may also force a 'modality switch', as evidenced by the whirligig beetle, *Dineutus sublineatus*, which has lost antennal lobes and instead has mushroom bodies supplied by the optic lobe, displaying a transition from olfactory to visual processing (*Lin and Strausfeld, 2012*). An alternative hypothesis would be that mushroom bodies in web-building species may integrate other sensory information, such as mechanosensation, relevant for web activities – which may also necessitate learning and memory processes, a well-established function of the MBs of insects such as *Drosophila* (*Modi et al., 2020*). Closer identification and annotation of the innervation patterns of non-visual sensory streams leading to the MBs would strengthen such a viewpoint.

While surprisingly, we did not observe co-expression for most of the neurosignaling immunostains within the MB, an exception was AstA immunoreactivity. AstA innervation of the MB has been reported in multiple insect species, although the extent of innervation throughout the MB components is variable (*Heuer et al., 2012*). Similarly to the general case in insects, AstA expression in *U. diversus* mushroom bodies appears to be supplied by extrinsic neurons, rather than the spider equivalent of the Kenyon cells, which would be found dorsal to the MB heads, and are not clearly immunoreactive. AstA function has so far been best understood in the context of peripheral and endocrine effects, but would likely act as a neuromodulator in the central nervous system (*Heuer et al., 2012*).

## Arcuate body

The arcuate body, being unmistakable and consistently present among species, is perhaps the best detailed structure in the spider brain, particularly in regard to innervation by neurotransmitter subtype populations. By aligning volumes to a common reference, the present methodology allowed for disambiguation of the layers innervated by specific signaling molecules and understanding of where these patterns overlap. In *U. diversus*, we confirmed two broad lobular divisions, which each contain an additional two major layers, supporting a number of structural motifs. Generalizing for the arcuate body innervation patterns in *U. diversus* of specific neuronal populations, as compared to *C. salei* and a few other species, one can conclude that there is a great degree of similarity in the relative arrangement of the gross layers, and even in certain fine structural details. This conclusion, extending to even the clustering of neurons giving rise to the layering patterns, has also been made in a study of CCAP immunoreactivity in the arcuate body of multiple arachnids (*Breidbach et al., 1995*). In comparative studies, the arcuate body has been found to compose a roughly proportionate percentage of the brain across the species examined, including web-builders and visually based hunters (*Napiórkowska and Kobak, 2018*; *Weltzien and Barth, 1991*), although this does not preclude potential differences in the density of synapses in this area. It is thus assuredly involved in various spider behaviors, and it will be illuminating to unravel how this conserved circuitry is harnessed for different ethological needs. The arcuate body lobes have been previously compared to the two nested neuropils known generally in insects as the upper and lower central bodies (fan-shaped body and ellipsoid body, respectively, in *Drosophila*) (*Homberg, 2008*; *Loesel et al., 2011*; *Strausfeld et al., 2006*) and the architecture of *U. diversus* supports these observations, showing obvious layering intersected by perpendicular neurites and columnar-like patterns.

To advance this argument, one might further compare the arcuate body layering of individual neurosignaling populations to the broader literature available for insect central complex, and indeed, commonalities are visible, such as GABAergic innervation primarily in only one of the two compartments of the central bodies (*Homberg et al., 2018*), multiple bands of serotonergic immunoreactivity in both compartments (*Homberg et al., 2023*), and the most prominent layer of dopaminergic signal being adjacent to the boundary between the two central bodies (*Timm et al., 2021*). Nevertheless, if attention is focused on differences, many distinctions could also be noted: FMRFamide (*Kaiser et al., 2022*) and proctolin innervation (*Kahsai and Winther, 2011*) of the central bodies of insects is more limited than the wide and multi-layered innervation in spiders, tyrosine hydroxylase immunoreactivity is also often in both central complex constituents (*Timm et al., 2021*), to name a few.

Recent comparative work in a large number of dicondylian insects using antibodies against GABAergic (*Homberg et al., 2018*), dopaminergic (*Timm et al., 2021*), and serotonergic (*Homberg et al., 2023*) populations is instructive for appreciating the variation in central body innervation which can be seen even between members of a clade. While immunostaining reliably describes signal in the central bodies and further attributes it to conserved neuronal types, there is additional diversity across

taxa in whether both upper and lower central bodies are innervated, as well as the form of ramification within the sublayers. Immunoreactivity in the PCB and noduli appears to be more sporadic, reported in a minority of the examined species.

Considering this variation, it seems prudent to be cautious in drawing parallels between layering patterns of specific neurosignaling molecules across a much wider evolutionary expanse, particularly when at present our knowledge of arcuate body patterning comes largely from only two species, *C. salei* and this study of *U. diversus*. Furthermore, for most of the antisera we used, we were unable to reliably trace the source of neuropil innervation to particular somata due to limited signal in processes, even in cases where the somata are clearly labeled. Given that more stringent arguments for homology appeal to the numbers, locations, and morphologies of supplying neurons, future studies would benefit from resolving these relationships to bolster the comparative argument.

## Novel neuropils

Structures which are conspicuous in our orb-building model spider but potentially not in hitherto studied cursorial species may be indicative of areas which are important for web-building. Nevertheless, it is not currently clear whether similar neuropils are absent in other species, or if they were simply not apparent by prior techniques. Apart from the mushroom bodies and arcuate body, neuropil structures within the interior of the protocerebrum have not been well distinguished. Multiple works refer to a 'central' or 'protocerebral neuropil' seemingly in regard to the undifferentiated mass of the protocerebrum as a whole. The image volume produced by aligning whole-mounted synganglia immunostained against synapsin and other targets instead reveals an intricacy of structures, beyond those described here. Two of the most conspicuous neuropils found in the dorsal supraesophageal mass are the PCB and the tonsillar neuropils.

Our description and multi-target staining of the PCB provides the clearest demonstration of such a structure in the spider to date. The use of this name has a precedent within the spider literature (*Seyfarth et al., 1993*), although whether the referent structure in *C. salei* is the same as in our model species will require additional clarification. Whether or not the authors chose this name in order to draw a parallel to the insect PCB is likewise ambiguous. A more promising candidate for this structure in past work is the 'posterior bridge', marked for *T. spinipalpis* and *M. muscosa*, seen adjacent to the arcuate body on the lateral–posterior edge of the central protocerebrum in acetylthiocholine iodide staining (*Meyer and Idel, 1977*). The PCB is a core constituent of the insect central complex (*Heinze, 2024*), but demonstrations in non-insect arthropods are scarcer. Examples have been found in crustaceans, such as the crayfish *Cherax destructor* (*Utting et al., 2000*), shore crab *H. sanguineus* (*Kotsyuba and Dyachuk, 2021*) as well as rock slater *Ligia occidentalis* and sidestriped shrimp *Pandalopsis dispar*, the latter of which shows widely arching, layered structure, stopping short of the midline (*Loesel et al., 2002*). We find such an anterior midline structure in *U. diversus*, possessing layers as revealed by antisera to neurotransmitter populations, and having a thinning (to absent) midline crossing, reminiscent of disjointed PCBs in certain insects including cockroaches and moths (https://insectbraindb.org/). A columnar pattern is not as of now forthcoming in the *U. diversus* PCB, which may be a consequence of density, as columnar structures can be difficult to see by immunohistochemistry (*Heinze, 2024*), demonstrated by the fact that the PCB is no more evidently columnar in cockroach than in the sidestriped shrimp when visualized with the same antisera to tachykinin-related peptide (*Loesel et al., 2002*).

A final undescribed neuropil which was apparent in the supraesophageal was the centrally located, tonsillar neuropil. Based on the ovoid form, paired appearance close to the midline, and close proximity to the unpaired midline neuropil(s) (arcuate body – ABv and Abd), the tonsillar neuropil bears a general resemblance to the noduli, a smaller constituent of the central complex of pterygote insects (*Heinze, 2024*). Among arthropods other than hexapods (including certain species of springtails; *Kollmann et al., 2011*), noduli have only thus far been found in stomatopods, where they were speculated to contribute to remarkable agility of the mantis shrimp (*Thoen et al., 2017*). Interestingly, as we have found serotonergic innervation to prominently define the tonsillar neuropil, 5-HT staining was also used to highlight domains within the noduli of mantis shrimp (*Thoen et al., 2017*), although the degree to which serotonergic innervation is present in insect noduli appears to be species-dependent (*Homberg et al., 2023*). Unlike the arcuate body and PCB, neither a columnar nor layered architecture is apparent in the tonsillar neuropil, although specific neurosignaling molecule stains concentrate in

certain domains, including a potential core and shell, as well as an anterior/posterior division. Noduli in insects also contain compartments, and the presence of layering is species-dependent (*Heinze, 2024*).

## A spider central complex?

Based on gross morphology, it is tempting to speculate that these novel neuropils, when considered along with each individual lobe of the arcuate body, may form an equivalent to a central complex in *U. diversus* (*Figure 13*). The central complex of insects is innervated and interconnected by tangential, columnar, and pontine neurons, forming a consistently identifiable relationship between neuropils across species (*Pfeiffer, 2023*). Apart from the crayfish (*Utting et al., 2000*), where neurons supplying the PCB also appear to innervate the central body, knowledge of intra-complex connectivity is lacking in non-insect arthropods. A detailed study of the Onychophoran (velvet worm, sister to arthropods) brain revealed several brain structures that appeared anatomically similar to those observed in arthropods (*Martin et al., 2022*). However, whether these ganglia are functionally homologous is a matter of debate. While the Onychophoran central body is more readily homologized to the arcuate body of Chelicerates, the homology of these structures to insect central bodies is not settled with arguments advanced for both views (*Strausfeld et al., 2006*; *Doeffinger et al., 2010*). The frontal body (which has gross similarities to the insect PCB) appears to lack columnar organization and lacks an obvious connection to the central body. No noduli were observed in the Onychophoran brain. The anatomy of this structure may be coincidental, or convergently evolved to execute functions relevant to the PCB.

Given that many of the antisera used in this study do not consistently trace neurites, the connectivity patterns between the neuropils of *U. diversus* supraesophageal mass require clarification. Future investigations employing techniques capable of isolating the ramification patterns of individual neurons within the context of the present neuropils in *U. diversus* will be essential to defining whether these currently disparate structures are truly members of a complex, and to what extent the connectivity is comparable to better studied arthropods. As a unit, the modules of the central complex integrate a variety of information including present orientation with respect to a salient environmental feature, memory of a heading goal, and speed – which can accomplish tasks such as path integration, migration, and other goal-directed movements relevant to particular species (*Honkanen et al., 2019*). While occurring in a much more spatially constrained context, these informational components could likewise be vital for organizing movements during the process of web-building, as well as maintaining a conception of the 360-degree web space as the spider strikes out to capture prey and subsequently return to the resting position at the hub. In such a scenario for *U. diversus* and other orb weavers, updates to present heading would likely be provided by mechanosensation, rather than optic flow, which has been shown to contribute even in insects which otherwise predominantly employ vision (*Turner-Evans et al., 2017*). The columnar segments of the central bodies maintain a representation of the flies' orientation within the environment in regard to a given feature (*Seelig and Jayaraman, 2015*). Although the exact number of columnar elements in the spider arcuate body lobes has not been established, they are numerous (with some suggestions in the thousands; *Homberg, 2008*), which could support a much more refined representation of the animal's radial self-made realm, underlying the often-stunning speed and precision with which the spider builds and navigates.

## Materials and methods

### Animals

Adult female *U. diversus* spiders were used for all neuroanatomical preparations. Spiders were housed freely in a greenhouse, or as 1–4 individuals in acrylic habitats within the lab, under 12:12 light–dark cycles.

### Immunohistochemistry

Spiders were anesthetized with carbon dioxide and rapidly dissected in HEPES-buffered saline (HBS) with 0.1% Triton X, and prepared for immunostaining following the methodology described by *Ott, 2008*. Samples were fixed overnight in ZnFA (0.25% $ZnCl_2$, 135 mM NaCl, 1.2% sucrose, 2% PFA) at 4°C. The following day, samples were washed 3 × 10 min in HBS + 0.1% Triton X on a nutator. Samples were dehydrated in 80% methanol/20% DMSO for 1 hr and 30 min, followed by 30 min in 100%

methanol. A series of 5-min incubations in 90%, 70%, 50%, 30%, and 0% methanol in 0.1 M Tris was applied, and the samples were blocked in 5% normal goat serum, 1% DMSO, in PBS with 0.1% Triton (PBST) for at least 1 hr. Primary antibodies were incubated for 3–5 days on a nutator at 4°C, before being washed with PBST for 3 × 15 min. Secondary antibodies were applied in blocking solution and incubated for 2–3 days on a nutator at 4°C. Secondary antibodies were washed off with 3 × 15 min washes with PBST, including DAPI (1:1000) in one of the wash steps. The sample was dehydrated for mounting through a series of 2%, 4%, 8%, 15%, 30%, 50%, 70%, and 80% glycerol in 0.1 M Tris for 20 min each. Nutation was performed for 2% through 15%, but only occasional hand agitation for the remaining steps. The sample was protected from light. Following 30 min of washing with 100% ethanol, most of the ethanol was pipetted off and the sample was underlaid with methyl salicylate and allowed to sink, where it was stored in the dark at room temperature until mounting.

For anti-TH staining, samples were dissected in Millonig's buffer with 0.1% Triton X, and fixed in 4% PFA in PBS for 45 min at room temperature while nutating. Immunostaining proceeded as described by *Auletta et al., 2020*. Samples were dehydrated and mounted in methyl salicylate by the steps used above for all other antibodies.

## Antibody characterization

The mouse synapsin antibody, anti-SYNORF1 (DHSB, 3C11, used at 1:100) is a monoclonal antibody raised against a purified GST–synapsin–GST fusion protein, and has been used extensively in invertebrates (*Ott, 2008*) and previously in spiders (*Steinhoff et al., 2017*). We also employed a rabbit polyclonal synapsin-I antibody (Abcam, ab64581, 1:500), which showed comparable staining to 3C11. We have found that recent stocks of this polyclonal antibody are not effective in *U. diversus*. A polyclonal chicken antibody raised against three synthetic peptides corresponding to human β-Tubulin3 (Aves Labs, TUJ, 1:250) was used for detection of tubulin. To recognize GABAergic neurons, we used a rabbit polyclonal anti-glutamic acid decarboxylase 65/67 (Sigma-Aldrich, G5163, 1:1000), made using the synthetic peptide KDIDFLIEEIERLGQDL of the C-terminal of GAD67. Dopaminergic populations were visualized with a mouse monoclonal antibody for tyrosine hydroxylase (ImmunoStar, 22941, 1:100), raised against purified tyrosine hydroxylase from rat PC12 cells. This antibody has been previously validated and employed in multiple spider species (*Auletta et al., 2020*). For cholinergic populations, we used a mouse monoclonal antibody against choline acetyltransferase (DHSB, ChAT4B1, 1:10) with confirmed species reactivity in arthropods. A rabbit polyclonal anti-serotonin antibody (ImmunoStar, 20080, 1:25) raised against serotonin-BSA and hence having species-independent reactivity, as previously used in spider (*Fabian and Seyfarth, 1997*), was used for serotonergic populations. An anti-tyrosine decarboxylase 2 antibody (CovaLab, pab0822-P, 1:250) raised against a synthetic peptide spanning a portion of the C-terminal of fruit fly TDC2 and recognizing the *D. melanogaster* 72 kDa TDC2 protein was used. For proctolin, a rabbit polyclonal anti-proctolin antibody was used (Biorbyt, orb122514, 1:250), having insect cross-reactivity. A mouse monoclonal antibody recognizing type A allatostatins (DHSB, 5F10, 1:10), raised against a synthetic AST7 peptide (APSGAQRLYGFGL) from cockroach and widely reactive across invertebrates was used for allatostatin A populations. A rabbit polyclonal anti-CCAP antibody was used which was raised against a synthetic peptide from *D. melanogaster* CCAP (Abcam, 58736, 1:250) and for FMRFamide, a rabbit polyclonal antibody (Sigma-Aldrich, ab15348, 1:25). Through the process of troubleshooting the whole-mount immunofluorescence protocol as well as testing antibodies, we produced multiple examples of samples which were negative for immunoreactivity, particularly in the interior of the brain. In these samples, we did not observe appreciable autofluorescence at any of the emission wavelengths used for visualizing the antibodies described above.

| Reagent | Host species | ID | Dilution |
|---|---|---|---|
| **Primary antibodies** | | | |
| 3C11 (anti SYNORF1) | Mouse | 3C11 (DHSB) RRID:AB_528479 | 1:100 |
| Anti-Synapsin I antibody – Synaptic Marker (ab64581) *polyclonal antibody – new stocks are not effective* | Rabbit | ab64581 (Abcam) | 1:500 |

*Continued on next page*

*Continued*

| Reagent | Host species | ID | Dilution |
|---|---|---|---|
| Anti-β-Tubulin3 | Chicken | TUJ (Aves Labs) | 1:250 |
| Anti-GAD | Rabbit | G5163 (Sigma-Aldrich) | 1:1000 |
| Anti-TH | Mouse | 22941 (ImmunoStar) | 1:100 |
| Anti-ChAT | Mouse | ChAT4B1 (DHSB) | 1:10 |
| Anti-5-HT | Rabbit | 20080 (ImmunoStar) | 1:25 |
| Anti-TDC2 | Rabbit | pab0822-P (CovaLab) | 1:250 |
| Anti-Proctolin | Rabbit | orb122514 | 1:250 |
| Anti-Allatostatin A | Mouse | 5F10-s (DHSB) | 1:10 |
| Anti-Cardioactive peptide | Rabbit | ab58736 (Abcam) | 1:250 |
| Anti-FMRFamide | Rabbit | ab15348 (Sigma-Aldrich) | 1:250 |
| Secondary antibodies | | | |
| 488 Goat Anti-Mouse | Goat | A-11001 (Thermo Fisher) | 1:500 |
| 555 Goat Anti-Rabbit | Goat | A21428 (Invitrogen) | 1:1000 |
| Alexa Fluor 647 Goat Anti-Chicken | Goat | 130-605-155 (Jackson) | 1:1000 |
| Alexa Fluor 647 Goat Anti-Horseradish Peroxidase (HRP) | Goat | 123-605-021 (Jackson) | 1:25 |

## Imaging

*U. diversus* synganglia were balanced upright, by placing samples subesophageal mass-side down in a well of methyl salicylate. Wells were constructed by adhering nested metal washers to a glass coverslip or slide using cyanoacrylate glue. A coverslip was also adhered to the top of the outer washer. Samples were imaged using a Zeiss LSM700 or LSM880 confocal microscope, with a LD LCI Plan-Apochromat 25x/0.8 Imm Corr DIC M27 objective (set to oil immersion), or a W Plan-Apochromat 20x/1.0 DIC D = 0.17 M27 75 mm water immersion objective, respectively. Full brain image volumes used for the standard brain were typically taken using a 1024 × 1024 image size, at 0.59 μm × 0.59 μm × 0.59 μm per pixel (with some earlier volumes imaged at 1.25 μm in the *z* dimension). Higher resolution stacks, such as for arcuate body layers, were imaged at 2048 × 2048 with 0.21 μm × 0.21 μm (*x,y*) and ~0.4–1.0 μm (*z*).

## Volume alignment

Alignment of confocal image volumes was performed using elastix 5.0.1 (*Klein et al., 2010*, *Shamonin, 2014*). Registration was performed first by a rigid method using an affine transform with an adaptive stochastic gradient descent optimizer for 20,000 iterations, with 40,000 spatial samples at 5 resolution levels. This was followed by a non-rigid registration using a B-spline transform with a standard gradient descent optimizer for 200,000 iterations at 5 resolution levels and using the AdvancedMattesMutualInformation metric. Transformation matrices were established using the anti-synapsin stain as a registration channel. A preliminary subset of synapsin volumes was mutually transformed onto each other, and the brain sample for which the most satisfactorily aligned pairings resulted was selected as the reference brain, onto which all other subsequent image volumes were aligned. The standard brain depicted in the figures above is an averaged composite of six aligned synapsin volumes. The final transformation matrix generated by registration of the synapsin channel was then applied to other channels present for each sample image volume (the neurosignaling immunostains).

In limited cases, no satisfactory image volume alignment could be obtained based on the elastix parameters specified previously. In these cases, we manually applied a small correction to the elastix output (the 'moving image') using radial basis function (RBF) interpolation. First, several location correspondences were manually annotated in the reference and moving image. An additional $N^3$

regularly spaced location correspondences were automatically created where no manual annotation was present within a 100-pixel distance, with $N$ = ceil[image axis length/100]. The moving image coordinates were subsequently transformed using RBF interpolation with a thin plate spline kernel.

## Visualization

Annotations of neuropils were drawn using ImageJ and Napari, and 3D renderings created using Imaris 10.1 (Oxford Instruments). Renderings of $z$-planes onto the 3D synapsin volume were created using VisPy (https://vispy.org/).

## Acknowledgements

GA acknowledges funding from the NSF Postdoctoral Research Fellowships in Biology Program under Grant No. 2109747. AC acknowledges funding from the Johns Hopkins Kavli Neuroscience Discovery Institute Doctoral Fellows Program. AG acknowledges funding from NIH (R35GM124883). The authors declare that they have no competing interests. All data needed to evaluate the conclusions in the paper are present in the paper and/or the Supplementary Materials. Raw data files can be accessed at 10.35077/ace-owl-gum.

## Additional information

### Funding

| Funder | Grant reference number | Author |
|---|---|---|
| National Institute of General Medical Sciences | R35GM124883 | Andrew Gordus |
| National Science Foundation | 2109747 | Gregory Artiushin |
| Kavli Neuroscience Discovery Institute Doctoral Fellows Program, Johns Hopkins University | | Abel Corver |

The funders had no role in study design, data collection, and interpretation, or the decision to submit the work for publication.

### Author contributions

Gregory Artiushin, Conceptualization, Resources, Data curation, Formal analysis, Funding acquisition, Investigation, Visualization, Methodology, Writing – original draft, Project administration, Writing – review and editing; Abel Corver, Software, Formal analysis, Methodology; Andrew Gordus, Conceptualization, Supervision, Funding acquisition, Project administration, Writing – review and editing

### Author ORCIDs

Gregory Artiushin ⓘ https://orcid.org/0000-0003-1615-3012
Andrew Gordus ⓘ https://orcid.org/0000-0002-5550-0286

Reviewer #1 (Public review): https://doi.org/10.7554/eLife.107732.3.sa1
Reviewer #2 (Public review): https://doi.org/10.7554/eLife.107732.3.sa2
Reviewer #3 (Public review): https://doi.org/10.7554/eLife.107732.3.sa3
Author response https://doi.org/10.7554/eLife.107732.3.sa4

## Additional files

### Supplementary files

MDAR checklist

## Data availability

TIFF stacks of volumes presented can be viewed and downloaded from the Brain Imaging Library (BIL): https://doi.org/10.35077/ace-owl-gum. Custom code used to align volumes with elastix can be found here: https://github.com/GordusLab/Artiushin-elastix-eLife (copy archived at *GordusLab, 2026*).

The following dataset was generated:

| Author(s) | Year | Dataset title | Dataset URL | Database and Identifier |
|---|---|---|---|---|
| Artiushin G, Corver A, Gordus A | 2025 | A three-dimensional immunofluorescence atlas of the brain of the hackled-orb weaver spider, Uloborus diversus | https://doi.org/10.35077/ace-owl-gum | Brain Image Library, 10.35077/ace-owl-gum |

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
