## [Editor Report · eLife Assessment]

This **valuable** study provides a 3D standardised anatomical atlas of the brain of an orb-weaving spider. The authors describe the brain's shape and its inner compartments-the neuropils-and add information on the distribution of a number of neuroactive substances such as neurotransmitters and neuropeptides. Through the use of histological and microscopy methods the authors provide a more complete view of an arachnid brain than previous studies and also presents **convincing** evidence about the organisation and homology of brain regions. The work will serve as a reference for future studies on spider brains and will enables comparisons of brain regions with insects so that the evolution of these structures can be inferred across arthropods.

---

## [Referee Report · Reviewer #1 (Public review)]

Summary:

Artiushin et al. establish a comprehensive 3D atlas of the brain of the orb-web building spider Uloborus diversus. First, they use immunohistochemistry detection of synapsin to mark and reconstruct the neuropils of the brain of six specimen and they generate a standard brain by averaging these brains. Onto this standard 3D brain, they plot immunohistochemical stainings of major transmitters to detect cholinergic, serotonergic, octopaminergic/taryminergic and GABAergic neurons, respectively. Further, they add information on the expression of a number of neuropeptides (Proctolin, AllatostatinA, CCAP and FMRFamide). Based on this data and 3D reconstructions, they extensively describe the morphology of the entire synganglion, the discernable neuropils and their neurotransmitter/neuromodulator content.

Strengths:

While 3D reconstruction of spider brains and the detection of some neuroactive substances have been published before, this seems to be the most comprehensive analysis so far both in terms of number of substances tested and the ambition to analyzing the entire synganglion. Interestingly, besides the previously described neuropils, they detect a novel brain structure, which they call the tonsillar neuropil.

Immunohistochemistry, imaging and 3D reconstruction are convincingly done and the data is extensively visualized in figures, schemes and very useful films, which allow the reader to work with the data. Due to its comprehensiveness, this dataset will be a valuable reference for researchers working on spider brains or on the evolution of arthropod brains.

Weaknesses:

As expected for such a descriptive groundwork, new insights or hypotheses are limited while the first description of the tonsillar neuropil is interesting. The reconstruction of the main tracts of the brain would be a very valuable complementary piece of data.

---

## [Referee Report · Reviewer #2 (Public review)]

Summary

Artiushin et al. created the first three-dimensional atlas of a synganglion in the hackled orb-weaver spider, which is becoming a popular model for web-building behavior. Immunohistochemical analysis with an impressive array of antisera reveal subcompartments of neuroanatomical structures described in other spider species as well as two previously undescribed arachnid structures, the protocerebral bridge, hagstone, and paired tonsillar neuropils. The authors describe the spider's neuroanatomy in detail and discuss similarities and differences from other spider species. The final section of the discussion examines the homology between onychophoran and chelicerate arcuate bodies and mandibulate central bodies.

Strengths

The authors set out to create a detailed 3D atlas and accomplished this goal.

Exceptional tissue clearing and imaging of the nervous system reveals the three-dimensional relationships between neuropils and some connectivity that would not be apparent in sectioned brains.

Detailed anatomical description makes it easy to reference structures described between the text and figures.

The authors used a large palette of antisera which may each be investigated in future studies for function in the spider nervous system and may be compared across species.

Weaknesses addressed in the revision

Additional added information about spider-specific neuropils helps orient a non-expert reader. While the function and connectivity of many of these structures is currently unknown, this study will be foundational in future investigations of function.

---

## [Referee Report · Reviewer #3 (Public review)]

Summary:

This is an impressive paper that offers a much-needed 3D standardized brain atlas for the hackled-orb weaving spider Uloborus diversus, an emerging organism of study in neuroethology. The authors used a detailed immunohistological wholemount staining method that allowed them to localize a wide range of common neurotransmitters and neuropeptides and map them on a common brain atlas. Through this approach, they discovered groups of cells that may form parts of neuropils that had not previously been described, such as the 'tonsillar neuropil', which might be part of a larger insect-like central complex. Further, this work provides unique insights into previously underappreciated complexity of higher-order neuropils in spiders, particularly the arcuate body, and hints at a potentially important role for the mushroom bodies in vibratory processing for web-building spiders.

Strengths:

To understand brain function, data from many experiments on brain structure must be compiled to serve as a reference and foundation for future work. As demonstrated by the overwhelming success in genetically tractable laboratory animals, 3D standardized brain atlases are invaluable tools-especially as increasing amounts of data are obtained at the gross morphological, synaptic, and genetic levels, and as functional data from electrophysiology and imaging are integrated. Among 'non-model' organisms, such approaches have included global silver staining and confocal microscopy, MRI, and more recently, micro-computed tomography (X-ray) scans used to image multiple brains and average them into a composite reference. In this study, the authors used synapsin immunoreactivity to generate an averaged spider brain as a scaffold for mapping immunoreactivity to other neuromodulators. Using this framework, they describe many previously known spider brain structures and also identify some previously undescribed regions. They argue that the arcuate body-a midline neuropil thought to have diverged evolutionarily from the insect central complex-shows structural similarities that may support its role in path integration and navigation.

Having diverged from insects such as the fruit fly *Drosophila melanogaster* over 400 million years ago, spiders are an important group for study-particularly due to their elegant web-building behavior, which is thought to have contributed to their remarkable evolutionary success. How such exquisitely complex behavior is supported by a relatively small brain remains unclear. A rich tradition of spider neuroanatomy emerged in the previous century through the work of comparative zoologists, who used reduced silver and Golgi stains to reveal remarkable detail about gross neuroanatomy. Yet, these techniques cannot uncover the brain's neurochemical landscape, highlighting the need for more modern approaches-such as those employed in the present study.

A key insight from this study involves two prominent higher-order neuropils of the protocerebrum: the arcuate body and the mushroom bodies. The authors show that the arcuate body has a more complex structure and lamination than previously recognized, suggesting it is insect central complex-like and may support functions such as path integration and navigation, which are critical during web building. They also report strong synapsin immunoreactivity in the mushroom bodies and speculate that these structures contribute to vibratory processing during sensory feedback, particularly in the context of web building and prey localization. These findings align with prior work that noted the complex architecture of both neuropils in spiders and their resemblance (and in some cases greater complexity) compared to their insect counterparts. Additionally, the authors describe previously unrecognized neuropils, such as the 'tonsillar neuropil,' whose function remains unknown but may belong to a larger central complex. The diverse patterns of neuromodulator immunoreactivity further suggest that plasticity plays a substantial role in central circuits.

Weaknesses:

My major concern, however, is some of the authors' neuroanatomical descriptions rely too heavily on inference rather than what is currently resolvable from their immunohistochemistry stains alone.

Comments on revisions:

I thought that the authors did an excellent job responding to the reviews, and I have no further comments.

---

## [Author Response]

The following is the authors’ response to the original reviews.

**Public Reviews:**

**Reviewer #1 (Public review):**
Summary:Artiushin et al. establish a comprehensive 3D atlas of the brain of the orb-web building spider Uloborus diversus. First, they use immunohistochemistry detection of synapsin to mark and reconstruct the neuropils of the brain of six specimens and they generate a standard brain by averaging these brains. Onto this standard 3D brain, they plot immunohistochemical stainings of major transmitters to detect cholinergic, serotonergic, octopaminergic/taryminergic and GABAergic neurons, respectively. Further, they add information on the expression of a number of neuropeptides (Proctolin, AllatostatinA, CCAP, and FMRFamide). Based on this data and 3D reconstructions, they extensively describe the morphology of the entire synganglion, the discernible neuropils, and their neurotransmitter/neuromodulator content.Strengths:While 3D reconstruction of spider brains and the detection of some neuroactive substances have been published before, this seems to be the most comprehensive analysis so far, both in terms of the number of substances tested and the ambition to analyze the entire synganglion. Interestingly, besides the previously described neuropils, they detect a novel brain structure, which they call the tonsillar neuropil.Immunohistochemistry, imaging, and 3D reconstruction are convincingly done, and the data are extensively visualized in figures, schemes, and very useful films, which allow the reader to work with the data. Due to its comprehensiveness, this dataset will be a valuable reference for researchers working on spider brains or on the evolution of arthropod brains.Weaknesses:As expected for such a descriptive groundwork, new insights or hypotheses are limited, apart from the first description of the tonsillar neuropil. A more comprehensive labeling in the panels of the mentioned structures would help to follow the descriptions. The reconstruction of the main tracts of the brain would be a very valuable complementary piece of data.
**Reviewer #2 (Public review):**
SummaryArtiushin et al. created the first three-dimensional atlas of a synganglion in the hackled orb-weaver spider, which is becoming a popular model for web-building behavior. Immunohistochemical analysis with an impressive array of antisera reveals subcompartments of neuroanatomical structures described in other spider species as well as two previously undescribed arachnid structures, the protocerebral bridge, hagstone, and paired tonsillar neuropils. The authors describe the spider's neuroanatomy in detail and discuss similarities and differences from other spider species. The final section of the discussion examines the homology between onychophoran and chelicerate arcuate bodies and mandibulate central bodies.StrengthsThe authors set out to create a detailed 3D atlas and accomplished this goal.Exceptional tissue clearing and imaging of the nervous system reveal the three-dimensional relationships between neuropils and some connectivity that would not be apparent in sectioned brains.A detailed anatomical description makes it easy to reference structures described between the text and figures.The authors used a large palette of antisera which may be investigated in future studies for function in the spider nervous system and may be compared across species.WeaknessesIt would be useful for non-specialists if the authors would introduce each neuropil with some orientation about its function or what kind of input/output it receives, if this is known for other species. Especially those structures that are not described in other arthropods, like the opisthosomal neuropil. Are there implications for neuroanatomical findings in this paper on the understanding of how web-building behaviors are mediated by the brain?Likewise, where possible, it would be helpful to have some discussion of the implications of certain neurotransmitters/neuropeptides being enriched in different areas. For example, GABA would signal areas of inhibitory connections, such as inhibitory input to mushroom bodies, as described in other arthropods. In the discussion section on relationships between spider and insect midline neuropils, are there similarities in expression patterns between those described here and in insects?
**Reviewer #3 (Public review):**
Summary:This is an impressive paper that offers a much-needed 3D standardized brain atlas for the hackled-orb weaving spider Uloborus diversus, an emerging organism of study in neuroethology. The authors used a detailed immunohistological whole-mount staining method that allowed them to localize a wide range of common neurotransmitters and neuropeptides and map them on a common brain atlas. Through this approach, they discovered groups of cells that may form parts of neuropils that had not previously been described, such as the 'tonsillar neuropil', which might be part of a larger insect-like central complex. Further, this work provides unique insights into the previously underappreciated complexity of higher-order neuropils in spiders, particularly the arcuate body, and hints at a potentially important role for the mushroom bodies in vibratory processing for web-building spiders.Strengths:To understand brain function, data from many experiments on brain structure must be compiled to serve as a reference and foundation for future work. As demonstrated by the overwhelming success in genetically tractable laboratory animals, 3D standardized brain atlases are invaluable tools - especially as increasing amounts of data are obtained at the gross morphological, synaptic, and genetic levels, and as functional data from electrophysiology and imaging are integrated. Among 'non-model' organisms, such approaches have included global silver staining and confocal microscopy, MRI, and, more recently, micro-computed tomography (X-ray) scans used to image multiple brains and average them into a composite reference. In this study, the authors used synapsin immunoreactivity to generate an averaged spider brain as a scaffold for mapping immunoreactivity to other neuromodulators. Using this framework, they describe many previously known spider brain structures and also identify some previously undescribed regions. They argue that the arcuate body - a midline neuropil thought to have diverged evolutionarily from the insect central complex - shows structural similarities that may support its role in path integration and navigation.Having diverged from insects such as the fruit fly *Drosophila melanogaster* over 400 million years ago, spiders are an important group for study - particularly due to their elegant web-building behavior, which is thought to have contributed to their remarkable evolutionary success. How such exquisitely complex behavior is supported by a relatively small brain remains unclear. A rich tradition of spider neuroanatomy emerged in the previous century through the work of comparative zoologists, who used reduced silver and Golgi stains to reveal remarkable detail about gross neuroanatomy. Yet, these techniques cannot uncover the brain's neurochemical landscape, highlighting the need for more modern approaches-such as those employed in the present study.A key insight from this study involves two prominent higher-order neuropils of the protocerebrum: the arcuate body and the mushroom bodies. The authors show that the arcuate body has a more complex structure and lamination than previously recognized, suggesting it is insect central complex-like and may support functions such as path integration and navigation, which are critical during web building. They also report strong synapsin immunoreactivity in the mushroom bodies and speculate that these structures contribute to vibratory processing during sensory feedback, particularly in the context of web building and prey localization. These findings align with prior work that noted the complex architecture of both neuropils in spiders and their resemblance (and in some cases greater complexity) compared to their insect counterparts. Additionally, the authors describe previously unrecognized neuropils, such as the 'tonsillar neuropil,' whose function remains unknown but may belong to a larger central complex. The diverse patterns of neuromodulator immunoreactivity further suggest that plasticity plays a substantial role in central circuits.Weaknesses:My major concern, however, is that some of the authors' neuroanatomical descriptions rely too heavily on inference rather than what is currently resolvable from their immunohistochemistry stains alone.

We would like to thank the reviewers for their time and effort in carefully reading our manuscript and providing helpful feedback, and particularly for their appreciation and realistic understanding of the scope of this study and its context within the existing spider neuroanatomical literature.

Regarding the limitations and potential additions to this study, we believe these to be well-reasoned and are in agreement. We plan to address some of these shortcomings in future publications.

As multiple reviewers remarked, a mapping of the major tracts of the brain would be a welcome addition to understanding the neuroanatomy of U. diversus. This is something which we are actively working on and hope to provide in a forthcoming publication. Given the length of this paper as is, we considered that a treatment of the tracts would be better served as an additional paper. Likewise, mapping of the immunoreactive somata of the currently investigated targets is a component which we would like to describe as part of a separate paper, keeping the focus of the current one on neuropils, in order to leverage our aligned volumes to describe co-expression patterns, which is not as useful for the more widely dispersed somata. Furthermore, while we often see somata through immunostaining, the presence and intensity of the signal is variable among immunoreactive populations. We are finding that these populations are more consistently and comprehensively revealed thru fluorescent in situ hybridization.

We appreciate the desire of the reviewers for further information regarding the connectivity and function of the described neuropils, and where possible we have added additional statements and references. That being said, where this context remains sparse is largely a reflection of the lack of information in the literature. This is particularly the case for functional roles for spider neuropils, especially higher order ones of the protocerebrum, which are essentially unexamined. As summarized in the quite recent update to Foelix’s Spider Neuroanatomy, a functional understanding for protocerebral neuropil is really only available for the visual pathway. Consequently, it is therefore also difficult to speak of the implications for presence or absence of particular signaling elements in these neuropils, if no further information about the circuitry or behavioral correlates are available. Finally, multiple reviewers suggested that it might be worthwhile to explore a comparison of the arcuate body layer innervation to that of the central bodies of insects, of which there is a richer literature. This is an idea which we were also initially attracted to, and have now added some lines to the discussion section. Our position on this is a cautious one, as a series of more recent comparative studies spanning many insect species using the same antibody, reveals a considerable amount of variation in central body layering even within this clade, which has given us pause in interpreting how substantive similarities and differences to the far more distant spiders would be. Still, this is an interesting avenue which merits an eventual comprehensive analysis, one which would certainly benefit from having additional examples from more spider species, in order to not overstate conclusions based on the currently limited neuroanatomical representation.

Given our framing for the impetus to advance neuroanatomical knowledge in orb-web builders, the question of whether the present findings inform the circuitry controlling web-building is one that naturally follows. While we are unable with this dataset alone to define which brain areas mediate web-building - something which would likely be beyond any anatomical dataset lacking complementary functional data – the process of assembling the atlas has revealed structures and defined innervation patterns in previously ambiguous sectors of the spider brain, particularly in the protocerebrum. A simplistic proposal is that such regions, which are more conspicuous by our techniques and in this model species, would be good candidates for further inquiries into web-building circuitry, as their absence or oversight in past work could be attributable to the different behavioral styles of those model species. Regardless, granted that such a hypothesis cannot be readily refuted by the existing neuroanatomical literature, underscores the need to have more finely refined models of the spider brain, to which we hope that we have positively contributed to and are gratified by the reviewer’s enthusiasm for the strengths of this study.

**Recommendations for the authors:**

**Reviewer #1 (Recommendations for the authors):**
(1) Brenneis 2022 has done a very nice and comprehensive study focused on the visual system - this might be worth including.

Thank you, we have included this reference on Line 34.

(2) L 29: When talking about "connectivity maps", the emerging connectomes based on EM data could be mentioned.

Additional references have been added, thank you. Line 35.

(3) L 99: Please mention that you are going to describe the brain from ventral to dorsal.

Thank you, we have added a comment to Line 99.

(4) L 13: is found at the posterior.

Thank you, revised.

(5) L 168: How did you pick those two proctolin+ somata, given that there is a lot of additional punctate signal?

Although not visible in this image, if you scroll through the stack there is a neurite which extends from these neurons directly to this area of pronounced immunoreactivity.

(6) Figure 1: Please add the names of the neuropils you go through afterwards.

We have added labels for neuropils which are recognizable externally.

(7) Figure 1 and Figure 5: Please mark the esophagus.

Label has now been added to Figure 1. In Figure 5, the esophagus should not really be visible because these planes are just ventral to its closure.

(8) Figure 5A: I did not see any CCAP signal where the arrow points to; same for 5B (ChAT).

In hindsight, the CCAP point is probably too minor to be worth mentioning, so we have removed it.

The ChAT signal pattern in 5B has been reinforced by adding a dashed circle to show its location as well.

(9) L 249: Could the circular spot also be a tract (many tracts lack synapsin - at least in insects)?

Yes, thank you for pointing this out – the sentence is revised (L274). We are currently further analyzing anti-tubulin volumes and it seem that indeed there are tracts which occupy these synapsin-negative spaces, although interestingly they do not tend to account for the entire space.

(10) L 302: Help me see the "conspicuous" thing.

Brace added to Fig. 8B, note in caption.

(11) L 315: Please first introduce the number of the eyes and how these relate to 1{degree sign} and 2{degree sign} pathway. Are these separate pathways from separate eyes or two relay stations of one visual pathway?

We have expanded the introduction to this section (L336). Yes, these are considered as two separate visual pathways, with a typical segregation of which eyes contribute to which pathway – although there is evidence for species-specific differences in these contributions. In the context of this atlas, we are not currently able to follow which eyes are innervating which pathway.

(12) L 343: It seems that the tonsillar neuropil could be midline spanning (at least this is how I interpret the signal across the midline). Would it make sense to re-formulate from a paired structure to midline-spanning? Would that make it another option for being a central complex homolog?

In the spectrum from totally midline spanning and unpaired (e.g., arcuate body (at least in adults)) to almost fully distinct and paired (e.g., mushroom bodies (although even here there is a midline spanning ‘bridge’)), we view the tonsillar to be more paired due to the oval components, although it does have a midline spanning section, particularly unambiguous just posterior to the oval sections.

Regarding central complex homology, if the suggestion is that the tonsillar with its midline spanning component could represent the entire central complex, then this is a possibility, but it would neglect the highly innervated and layered arcuate body, which we think represent a stronger contender – at least as a component of the central complex. For this reason, we would still be partial to the possibility that the tonsillar is a part of the central complex, but not the entire complex.

(13) L 407: ...and dorsal (..) lobe...

Added the word ‘lobe’ to this sentence (L429).

(14) L 620ff: Maybe mention the role of MBs in learning and memory.

A reference has been added at L661.

(15) L 644: In the context of arcuate body homology with the central body, I was missing a discussion of the neurotransmitters expressed in the respective parts in insects. Would that provide additional arguments?

This is an interesting comparison to explore, and is one that we initially considered making as well. There are certainly commonalities that one could point to, particularly in trying to build the case of whether particular lobes of the arcuate body are similar to the fan-shaped or ellipsoid bodies in insects. Nevertheless, something which has given us pause is studying the more recent comparative works between insect species (Timm et al., 2021, J Comp Neuro, Homberg et al., 2023, J Comp Neuro), which also reveal a fair degree of heterogeneity in expression patterns between species – and this is despite the fact that the neuropils are unambiguously homologous. When comparing to a much more evolutionarily distant organism such as the spider, it becomes less clear which extant species should serve as the best point of comparison, and therefore we fear making specious arguments by focusing on similarities when there are also many differences. We have added some of these comments to the discussion (L699-725).

Throughout the text, I frequently had difficulties in finding the panels right away in the structures mentioned in the text. It would help to number the panels (e.g., 6Ai, Aii, Aii,i etc) and refer to those in the text. Further, all structures mentioned in the text should be labelled with arrows/arrowheads unless they are unequivocally identified in the panel

Thank you for the suggestion. We have adopted the additional numbering scheme for panels, and added additional markers where suggested.

**Reviewer #2 (Recommendations for the authors):**
(1) L 18: "neurotransmitter" should be pluralized.

Thank you, revised (L18).

(2) L 55: Missing the word "the" before "U. diversus".

Thank you, revised (L57).

(3) L 179: Change synaptic dense to "synapse-dense".

Thank you, revised (L189).

(4) L 570: "present in" would be clearer than "presented on in".

Our intention here was to say that Loesel et al did not show slices from the subesophageal mass for CCAP, so it was ambiguous as to whether it had immunoreactivity there but they simply did not present it, or if it indeed doesn’t show signal in the subesophageal. But agreed, this is awkward phrasing which has been revised (L606-608), thank you.

(5) L 641: It would be worth noting that the upper and lower central bodies are referred to as the fan-shaped and ellipsoid bodies in many insects.

Thank you, this has been added in L694.

(6) L 642: Although cited here regarding insect central body layers, Strausfeld et al. 2006 mainly describe the onychophoran brain and the evolutionary relationship between the onychophoran and chelicerate arcuate bodies. The phylogenetic relationships described here would strengthen the discussion in the section titled "A spider central complex?"

The phylogenetic relationship of onychophorans and chelicerates remains controversial and therefore we find it tricky to use this point to advance the argument in that discussion section, as one could make opposing arguments. The homology of the arcuate body (between chelicerates, onychophorans, and mandibulates) has likewise been argued over, with this Strausfeld et al paper offering one perspective, while others are more permissive (good summary at end of Doeffinger et al., 2010). Our thought was simply to draw attention to grossly similar protocerebral neuropils in examples from distantly related arthropods, without taking a stance, as our data doesn’t really deeply advance one view over the other.

(7) L 701- Noduli have been described in stomatopods (Thoen et al., Front. Behav. Neurosci., 2017).

This is an important addition, thank you – it has been incorporated and cited (L766).

(8) Antisera against DC0 (PKA-C alpha) may distinguish globuli cells from other soma surrounding the mushroom bodies, but this may be accomplished in future studies.

Agreed, this is something we have been interested in, but have not yet acquired the antibody.

**Reviewer #3 (Recommendations for the authors):**
Overall, this paper is both timely and important. However, it may face some resistance from classically trained arthropod neuroanatomists due to the authors' reliance on immunohistochemistry alone. A method to visualize fiber tracts and neuropil morphology would have been a valuable and grounding complement to the dataset and can be added in future publications. Tract-tracing methods (e.g., dextran injections) would strengthen certain claims about connectivity - particularly those concerning the mushroom bodies. For delineating putative cell populations across regions, fluorescence in situ hybridization for key transcripts would offer convincing evidence, especially in the context of the arcuate body, the tonsillar neuropil, and proposed homologies to the insect central complex.That said, the dataset remains rich and valuable. Outlined below are a number of issues the authors may wish to address. Most are relatively minor, but a few require further clarification.(1) Abstract(a) L 12-14: The authors should frame their work as a novel contribution to our understanding of the spider brain, rather than solely as a tool or stepping stone for future studies. The opening sentences currently undersell the significance of the study.

Thank you for your encourament! We have revised the abstract.

(b) Rather than touting "first of its kind" in the abstract, state what was learned from this.

Thank you, we have revised the abstract.

(c) The abstract does not mention the major results of the study. It should state which brain regions were found. It should list all of the peptides and transmitters that were tested so that they can be discoverable in searches.

Thank you, revised.

(2) Introduction(a) L 38: There's a more updated reference for Long (2016): Long, S. M. (2021). Variations on a theme: Morphological variation in the secondary eye visual pathway across the order of Araneae. Journal of Comparative Neurology, 529(2), 259-280.

Thank you, this has been updated (L41 and elsewhere).

(b) L 47: While whole-mount imaging offers some benefits, a downside is the need for complete brain dissection from the cuticle, which in spiders likely damages superficial structures (such as the secondary eye pathways).

True – we have added this caveat to the section (L48-51).

(c) L 49-52: If making this claim, more explicit comparisons with non-web building C. saeli in terms of neuropil presence, volume, or density later in the paper would be useful.

We do not have the data on hand to make measured comparisons of C. salei structures, and the neuropils identified in this study are not clearly identifiable in the slices provided in the literature, so would likely require new sample preparations. We’ve removed the reference to proportionality and softened this sentence slightly – we are not trying to make a strong claim, but simply state that this is a possibility.

(3) Results(a) The authors should state how they accounted for autofluorescence.

While we did not explicitly test for autofluorescence, the long process of establishing a working whole-mount immuno protocol and testing antibodies produced many examples of treated brains which did not show any substantial signal. We have added a note to the methods section (L866).

(b) L 69: There is some controversy in delineating the subesophageal and supraesophageal mass as the two major divisions despite its ubiquity in the literature. It might be safer to delineate the protocerebrum, deutocerebrum, and fused postoral ganglia (including the pedipalp ganglion) instead.

Thank you for this insight, we have modified the section, section headings and Figure 1 to account for this delineation as well. We have chosen to include both ways of describing the synganglion, in order to maintain a parallel with the past literature, and to be further accessible to non-specialist readers. L73-77

(c) L 90: It might be useful to include a justification for the use of these particular neuropeptides.

Thank you, revised. L97-99.

(d) L 106 - 108: It is stated that the innervation pattern of the leg neuropils is generally consistent, but from Figure 2, it seems that there are differences. The density of 5HT, Proctolin, ChAT, and FMRFamide seems to be higher in the posterior legs. AstA seems to have a broader distribution in L1 and is absent in L4.

We would still stand by the generalization that the innervation pattern is fairly similar for each leg. The L1 neuropils tend to be bigger than the posterior legs, which might explain the difference in density. Another important aspect to keep in mind is that not all of the leg neuropils appear at the exact same imaging plane as we move from ventral to dorsal. If you scroll through the synapsin stack (ventral to dorsal), you will see that L2 and L3 appear first, followed shortly by L1, and then L4, and at the dorsal end of the subesophageal they disappear in the opposite order. The observations listed here are true for the single z-plane in Figure 2, but the fact that they don’t appear at the same time seems to mainly account for these differences. For example, if you scroll further ventrally in the AstA volume, you will see a very similar innervation appear in L4 as well, even though it is absent in the Fig. 2 plane. We plan to have these individual volumes available from a repository so that they can be individually examined to better see the signal at all levels. At the moment, the entire repository can be accessed here: https://doi.org/10.35077/ace-moo-far.

(e) Figure 1 and elsewhere: The axes for the posterior and lateral views show Lateral and Medial. It would be more accurate to label them Left and Right. because it does not define the medial-to-lateral axis. The medial direction is correct for only one hemiganglion, and it's the opposite for the contralateral side.

Thank you, revised.

(f) In Figures that show particular sections, it might be helpful to include a plane in the standard brain to illustrate where that section is.

Yes, we agree and it was our original intention. It is something we can attempt to do, but there is not much room in the corners of many of the synapsin panels, making it harder to make the 3D representation big enough to be clear.

(g) Figure 2, 3: Presenting the z-section stack separately in B and C is awkward because it makes it seem that they are unrelated. I think it would be better to display the z160-190 directly above its corresponding z230-260 for each of the exemplars in B and C. Since there's no left-right asymmetry, a hemibrain could be shown for all examples as was done for TH in D. It's not clear why TH was presented differently.

Thank you for this suggestion. We rearranged the figure as described, but ultimately still found the original layout to be preferrable, in part because the labelling becomes too cramped. We hope that the potential confusion of the continuity of the B and C sections will be mitigated by focusing on the z plane labels and overall shape – which should suggest that the planes are not far from each other. We trust that the form of the leg neuropils is recognizable in both B and C synapsin images, and so readers will make the connection.

Regarding TH, this panel is apart from the rest because we were unable to register the TH volume to the standard brain because the variant of the protocol which produced good anti-TH staining conflicted with synapsin, and we could not simultaneously have adequate penetration of the synapsin signal. We did not want to align the TH panel with the others to avoid potential confusion that this was a view from the same z-plane of a registered volume, as the others are. We have added a note to the figure caption.

(h) The locations of the labels should be consistent. The antisera are below the images in Figure 2, above in Figure 3, and to the bottom left in Figure 5. The slices are shown above in Figure 2 and below in Figure 3.

Thank you, this has been revised for better consistency.

(i) It is surprising to me that there is no mention of the neuronal somata visible in Figure 2 and Figure 3. A typical mapping of the brain would map the locations of the neurons, not just the neuropils.

Our first arrangement of this paper described each immunostain individually from ventral to dorsal, including locations of the immunoreactive somata which could be observed. To aid the flow of the paper and leverage the aligned volumes to emphasize co-expression in the function divisions of the brain, we re-formulated to this current layout which is organized around neuropils. Somata locations are tricky to incorporate in this format of the paper which focuses on key z-planes or tight max projections, because the relevant immunoreactive somata are more dispersed throughout the synganglion, not always overlapping in neighboring z-planes. Further, since only a minority of the antisera we used can reveal traceable projections from the supplying somata in the whole-mount preparation, we would be quite limited in the degree to which we could integrate the specific somata mapping with expression patterns in the neuropil. Finally, compared to immuno, which can be variable in staining intensity between somata for the same target, we find that FISH reveals these locations more clearly and comprehensively – so while we agree that this mapping would also be useful for the atlas, we would like to better provide this information in a future publication using whole-mount FISH.

(j) L 139: There is a reference to a "brace" in Figure 3B, which does not seem to exist. There's one in Figure 3C.

There is a smaller brace near the bottom of the TDC2 panel in Fig. 3B.

(k) L 151 should be "3D".

Thank you, revised (L160).

(l) Figure 4C: It is not mentioned in the legend that the bottom inset is Proctolin without synapsin.

Thank you, revised (L1213).

(m) L 199: Are the authors sure this subdivision is solely on the anterior-posterior axis? Could it also be dorsal ventral? (i.e., could this be an artifact of the protocerebrum and deutocerebrum?)

Yes, this division can be appreciated to extend somewhat in the dorsal-ventral axis and it is possible that this is the protocerebrum emerging after the deutocerebrum, although this area is largely dorsal to the obvious part of the deutocerebrum. In the horizontal planes there appears to be a boundary line which we use for this subdivision in order to assist in better describing features within this generally ventral part of the protocerebrum – referred to as “stalk” because it is thinner before the protocerebrum expands in size, dorsally. Our intention was more organizational, and as stated in the text, this area is likely heterogenous and we are not suggesting that it has a unified function, so being a visual artifact would not be excluded.

(n) L 249: Could it also indicate large tracts projecting elsewhere?

Yes, definitely, we have evidence that part of the space is occupied by tracts. Revised, thank you (L262).

(o) L 281: Several investigators, including Long (2021,) noted very large and robust mushroom bodies of Nephila.

Thank you – the point is well taken that there are examples of orb-web builders that do have appreciable mushroom bodies. We have added a note in this section (L295), giving the examples of Deinopis spinosa and Argiope trifasciata (Figure 4.20 and 4.22 in Long, 2016).

It looks like these species make the point better than Nephila, as Long lists the mushroom body percentage of total protocerebral volume for D. spinosa as 4.18%, for A. trifasciata as 2.38%, but doesn’t give a percentage for Nephila clavipes (Figure 4.24) and only labels the mushroom bodies structures as “possible” in the figure.In Long (2021), Nephilidae is described as follows: “In Nephilidae, I found what could be greatly reduced medullae at the caudal end of the laminae, as well as a structure that has many physical hallmarks of reduced mushroom bodies”(p) L 324: If the authors were able to stain for histamine or supplement this work with a different dissection technique for the dorsal structures, the visual pathways might have been apparent, which seems like a very important set of neuropils to include in a complete brain atlas.

Yes, for this reason histamine has been an interesting target which we have attempted to visualize, but unfortunately have not yet been able to successfully stain for in U. diversus. An additional complication is that the antibodies we have seen call for glutaraldehyde fixation, which may make them incompatible with our approach to producing robust synapsin staining throughout the brain.

We agree that the lack of the complete visual pathway is a substantial weakness of our preparation, and should be amended in future work, but this will likely require developing a modified approach in order to preserve these delicate structures in U. diversus.

(q) L 331: Is this bulbous shape neuropil, or just the remains of neuropil that were not fully torn away during dissection?

This certainly is a severed part of the primary pathway, although it seems more likely that the bulbous shape is indicative of a neuropil form, rather than just being a happenstance shape that occurred during the breakage. We have examples where the same bulbous shape appears on both sides, and in different brains. It is possible that this may be the principal eye lamina – although we did not see co-staining with expected markers in examples where it did appear, so cannot be sure.

(r) L 354: Is tyraminergic co-staining with the protocerebral bridge enough evidence to speculate that inputs are being supplied?

We agree that this is not compelling, and have removed the statement.

(s) L 372: This whole structure appears to be a previously described structure in spiders, the 'protocerebral commissure'.

We are reasonably sure that what we are calling the PCB is a distinct structure from the protocerebral bridge (PCC). In Babu and Barth’s (1984) horizontal slice (Fig. 11b), you can see the protocerebral commissure immediately adjacent to the mushroom body bridge. It is found similarly located in other species, as can be seen in the supplementary 3D files provided by Steinhoff et al., (2024).

While not visible with synapsin in U. diversus, we likewise can make out a commissure in this area in close proximity to the mushroom body bridge using tubulin staining. What we are calling the protocerebral bridge is a structure which is much more dorsal to the protocerebral commissure, not appearing in the same planes as the MB bridge.

(t) L 377: Do you have an intuition why the tonsillar neuropil and the protocerebral bridge would show limited immunoreactivity, while the arcuate body's is quite extensive?

This is an interesting question. Given the degree of interconnection and the fact that multiple classes of neurons in insects will innervate both central body as well as PCB or noduli, perhaps it would be expected that expression in tonsillar and protocerebral bridge should be commensurate to the innervation by that particular neurotransmitter expressing population in the arcuate body. Apart from the fact that the arcuate body is just bigger, perhaps this points to a great role of the arcuate body for integration, whereas the tonsillar and PCB may engage in more particular processing, or be limited to certain sensory modalities.

Interestingly, it seems that this pattern of more limited immunoreactivity in the PCB and noduli compared with the central bodies (fan-shaped/ellipsoid) also appears in insects (Kahsai et al., 2010, J Comp Neuro, Timm et al., 2021, J Comp Neuro, Homberg et al., 2023, J Comp Neuro) – particularly, with almost every target having at least some layering in the fan-shaped body (Kahsai et al., 2010, J Comp Neuro). For example, serotoninergic innervation is fairly consistently seen in the upper and lower central bodies across insects, but its presence in the PCB or noduli is more variable – appearing in one or the other in a species-dependent manner (Homberg et al., 2023, J Comp Neuro).

(4) Discussion(a) L 556: But if confocal images from slices are aligned, is the 3D shape not preserved?

Yes, fair enough – the point we wanted to make was that there is still a limitation in z resolution depending on the thickness of the slices used, which could obscure structures, but perhaps this is too minor of a comment.

(b) L 597: This is a very interesting result. I agree it's likely to do with the processing of mechanosensory information relevant to web activities, and the mushroom body seems like the perfect candidate for this.(c) L 638: Worth noting that neuropil volume vs density of synapses might play a role in this, as the literature is currently a bit ambiguous with regards to the former.

Thank you, noted (L689).

(d) L 651: The latter seems far more plausible.

Agreed, though the presence of mushroom bodies appears to be variable in spiders, so we didn’t want to take a strong stance, here.